# Straggler-Resilient Decentralized Learning via Adaptive Asynchronous Updates

## Abstract

With the increasing demand for large-scale training of machine learning models, fully decentralized optimization methods have recently been advocated as alternatives to the popular parameter server framework. In this paradigm, each worker maintains a local estimate of the optimal parameter vector, and iteratively updates it by waiting and averaging all estimates obtained from its neighbors, and then corrects it on the basis of its local dataset. However, the synchronization phase is sensitive to stragglers. An efficient way to mitigate this effect is to consider asynchronous updates, where each worker computes stochastic gradients and communicates with other workers at its own pace. Unfortunately, fully asynchronous updates suffer from staleness of the stragglers' parameters. To address these limitations, we propose a fully decentralized algorithm `DSGD-AAU` with adaptive asynchronous updates via adaptively determining the number of neighbor workers for each worker to communicate with. We show that `DSGD-AAU` achieves a linear speedup for convergence (i.e., convergence performance increases linearly with respect to the number of workers). Experimental results on a suite of datasets and deep neural network models are provided to verify our theoretical results.

## 1 Introduction

Highly over-parameterized deep neural networks (DNNs) have shown impressive results in a variety of machine learning (ML) tasks such as computer vision He et al. (2016), natural language processing Vaswani et al. (2017) and many others. Its success depends on the availability of large amount of training data, which often leads to a dramatic increase in the size, complexity, and computational power of the training systems. In response to these challenges, the need for efficient parallel and distributed algorithms (i.e., data-parallel mini-batch stochastic gradient descent (SGD)) becomes even more urgent for solving large scale optimization and ML problems Bekkerman et al. (2011); Boyd et al. (2011). These ML algorithms in general use the parameter server (PS) Valiant (1990); Smola & Narayanamurthy (2010); Dean et al. (2012); Ho et al. (2013); Li et al. (2014a;b) or ring All-Reduce Gibiansky (2017); Goyal et al. (2017) communication primitive to perform exact averaging on the local mini-batch gradients computed on different subsets of the data by each worker, for the later synchronized model update. However, the aggregation with PS or All-Reduce often leads to extremely high communication overhead, causing the bottleneck in efficient ML model training. Additionally, since all workers need to communicate with the server concurrently, the PS based framework suffers from the single point failure and scalability issues, especially in large-scale cloud computing systems Lian et al. (2017).

Recently, an alternative approach has been developed in ML community Lian et al. (2017; 2018), where each worker keeps updating a local version of the parameter vector and broadcasts its updates *only* to its neighbors. This family of algorithms became originally popular in the control community, starting from the seminal work of Tsitsiklis et al. (1986) on distributed gradient methods, in which, they are referred to as *consensus-based distributed optimization methods* or *fully decentralized learning* (DSGD)[1] Neglia et al. (2020; 2019). The key idea of DSGD is to replace communication with the PS by peer-to-peer communication between individual workers.

---

[1]We interchangeably use "consensus-based methods" and "fully decentralized learning" (DSGD) in the rest of the paper.

Most existing consensus-based algorithms assume synchronous updates, where all workers perform SGD at the same rate in each iteration. Such a *synchronous update* is known to be sensitive to *stragglers*, i.e., slow tasks, because of the synchronization barrier at each iteration among *all workers*. The straggler issues are especially pronounced in heterogeneous computing environments such as cloud computing, where the computing speed of workers often varies significantly. In practice, only a small fraction of workers participate in each training round, thus rendering the active worker (neighbor) set stochastic and time-varying across training rounds. An efficient way to mitigate the effect of stragglers is to perform *fully asynchronous SGD*, where each worker updates its local parameter *immediately* once it completes the local SGD and moves to the next iteration *without waiting for any neighboring workers* for synchronization Lian et al. (2018); Luo et al. (2020); Assran & Rabbat (2020); Wang et al. (2022). Unfortunately, such fully asynchronous SGD often suffers from staleness of the stragglers' parameters and heavy communication overhead, and requires deadlock-avoidance (See Section 3). This raises the question: *Is it possible to get the best of both worlds of synchronous SGD and fully asynchronous SGD in fully decentralized (consensus-based) learning?*

In this paper, we advocate *adaptive asynchronous updates* (AAU) in fully decentralized learning that is theoretically justified to keep the advantages of both synchronous SGD and fully asynchronous SGD. Specifically, in each iteration, only a subset of workers participate (rather than all as in synchronous SGD for decentralized learning), and each worker only waits for a subset of neighbors to perform local model updates (instead of waiting for no neighbor as in fully asynchronous SGD for decentralized learning). On the one hand, these participated workers are the fastest computational workers, which addresses the straggler issues in synchronous SGD. On the other hand, each worker does not need to perform local update immediately using stale information or communicate with all neighbors, which addresses the stateless and heavy-communication issues in fully asynchronous SGD. We make the following contributions:

• We formulate the fully decentralized (consensus-based) optimization problem with AAU, and propose a decentralized SGD with AAU (`DSGD-AAU`) algorithm that adaptively determines the number of neighbor workers for each worker to perform local model update during the training time.

• To our best knowledge, we present the first convergence analysis of fully decentralized learning with AAU that is cognizant of training process at each worker. We show that the convergence rate of `DSGD-AAU` on both independent-identically-distributed (i.i.d.) and non-i.i.d. datasets across workers is $\mathcal{O}\left(\frac{1}{\sqrt{NK}}\right)$, where $N$ is the total number of workers and $K$ is the total number of communication rounds. This indicates that `DSGD-AAU` achieves a linear speedup of communication w.r.t. $N$ for a sufficiently large $K$. The state-of-the-art PSGD Lian et al. (2017) and AD-PSGD Lian et al. (2018) also achieve the same rate with synchronous updates or asynchronous updates with two randomly selected workers in each round, which lead to high communication costs and implementation complexity. In contrast we have the flexibility to *adaptively* determine the number of neighbor workers for each worker in the system. In addition, our proof does not require a bounded gradient assumption.

• We conduct experiments on image classification (CIFAR-10, MNIST, Tiny-ImageNet) and next-character prediction (Shakespeare) tasks with several representative DNN models, demonstrating that `DSGD-AAU` substantially accelerates training of DNNs by mitigating the effect of stragglers and consistently outperforms state-of-the-art methods.

**Notation.** Let $N$ be the total number of workers and $K$ be the number of total communication rounds. We denote the cardinality of a finite set $\mathcal{A}$ as $|\mathcal{A}|$. We denote by $\mathbf{I}_M$ and $\mathbf{1}$ ($\mathbf{0}$) the identity matrix and all-one (zero) matrices of proper dimensions, respectively. We also use $[N]$ to denote the set of integers $\{1, \cdots, N\}$. We use boldface to denote matrices and vectors, and $\|\cdot\|$ to denote the $l_2$-norm.

## 2    Background

In this section, we provide a brief overview of consensus-based decentralized optimization problem. We formulate the general decentralized ML problem with $N$ independent workers into the following optimization

task

$$\min_{\mathbf{w}} \ F(\mathbf{w}) := \frac{1}{N} \sum_{j=1}^{N} F_j(\mathbf{w}), \tag{1}$$

which aims to optimize a set of parameters $\mathbf{w} \in \mathbb{R}^{d \times 1}$ using $L$ examples from local training datasets $\mathcal{D}_j = \{\mathbf{x}_\ell, \ell \in [L]\}, \forall j \in [N]$. In particular, $F_j(\mathbf{w}) = \frac{1}{L} \sum_{\mathbf{x}_\ell \in \mathcal{D}_j} f(\mathbf{w}, \mathbf{x}_\ell)$ with $f(\mathbf{w}, \mathbf{x}_\ell)$ is the model error on the $\ell$-th element of dataset $\mathcal{D}_j$ when parameter $\mathbf{w}$ is used. In conventional decentralized learning, the distribution for each worker's local dataset is usually assumed to be i.i.d.. Unfortunately, this assumption may not hold true in practice since data can be generated locally by workers based on their circumstances. Instead, we make no such assumption in our model and our analysis holds for both i.i.d. and non-i.i.d. local datasets across workers. See Section 4 for details.

The decentralized system can be modeled as *a communication graph* $\mathcal{G} = (\mathcal{N}, \mathcal{E})$ with $\mathcal{N} = [N]$ being the set of workers and an edge $(i, j) \in \mathcal{E}$ indicates that workers $i$ and $j$ can communicate with each other. We assume that the graph is strongly connected Nedic & Ozdaglar (2009); Nedić et al. (2018), i.e., there exists at least one path between any two arbitrary workers. Denote neighbors of worker $j$ as $\mathcal{N}_j = \{i | (i, j) \in \mathcal{E}\} \cup \{j\}$. All workers perform local model updates synchronously. Specifically, worker $j$ maintains a local estimate of the parameter vector $\mathbf{w}_j(k)$ at iteration $k$ and broadcasts it to its neighbors. The local estimate is updated as follows:

$$\mathbf{w}_j(k+1) = \sum_{i \in \mathcal{N}_j} \left[ \mathbf{w}_i(k) - \eta g_j(\mathbf{w}_j(k)) \right] P_{i,j}, \tag{2}$$

where $\mathbf{P} = (P_{i,j})$ is a $N \times N$ non-negative matrix and we call it the *consensus matrix*, and $\eta > 0$ is the learning rate. In other words, in each iteration, worker $j \in [N]$ computes a weighted average (i.e., consensus component) of the estimates of its neighbors and itself, and then corrects it by taking into account a stochastic subgradient $g_j(\mathbf{w}_j(k))$ of its local function, i.e.,

$$g_j(\mathbf{w}_j(k)) := \frac{1}{|\mathcal{C}_j(k)|} \sum_{\mathbf{x}_\ell, \in \mathcal{C}_j(k)} \nabla f(\mathbf{w}_j(k), \mathbf{x}_\ell), \tag{3}$$

where $\mathcal{C}_j(k)$ is a random subset of $\mathcal{D}_j$ at iteration $k$.

## 3 `DSGD-AAU`: Decentralized SGD with Adaptive Asynchronous Updates

The performance of synchronous updates in (2) may be significantly dragged down by stragglers (i.e., workers with slow computational capabilities), since every worker $j$ needs to wait for all neighbors $i \in \mathcal{N}_j$ to complete the current local gradient computation $g_i(\mathbf{w}_i(k))$ at each iteration $k$ Lian et al. (2017). In other words, *in each synchronous iteration, all workers participate, and each worker needs to wait for all its neighbors*, as illustrated in Figure 1a.

To mitigate the effect of stragglers in synchronous updates in (2), a possible way is to perform fully asynchronous updates, where each worker $j$ updates the local parameter $\mathbf{w}_j$ immediately once it completes the local gradient computation and moves to the next iteration Lian et al. (2018); Luo et al. (2020); Assran & Rabbat (2020); Wang et al. (2022). In other words, *in each asynchronous iteration, only one (or more) worker participates, and each worker does not wait for any neighbors*, as illustrated in Figure 1b.

For instance, the worker randomly selects a neighbor to exchange local parameters once it completes the local gradient computation Lian et al. (2018), or the worker maintains a buffer to store information from neighbors whenever they complete a local update, and conducts a consensus update with the stale information in the buffer Assran & Rabbat (2020). As a result, both approaches suffer from staleness of the stragglers' parameters, as illustrated in Figure 1b. Exacerbating the problem is the fact that the method in Lian et al. (2018) only works for bipartite communication graph due to the deadlock issue, and the method in Assran & Rabbat (2020) suffers from heavy communications and memory space.

To address the aforementioned limitations, we advocate *adaptive asynchronous updates* (AAU) to get *the best of both worlds* of synchronous updates and fully asynchronous updates. Specifically, in each asynchronous

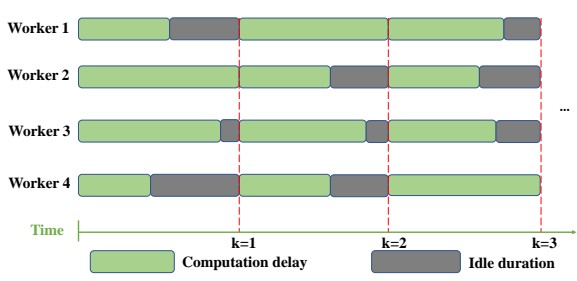

(a) Decentralized SGD with synchronous updates.

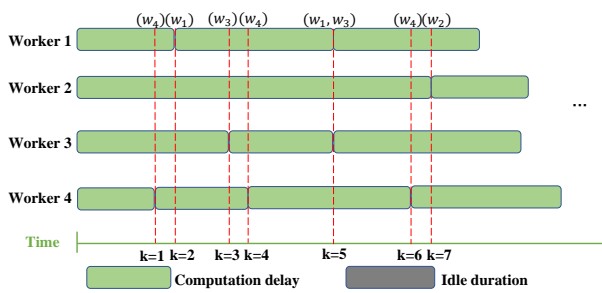

(b) Decentralized SGD with asynchronous updates.

Figure 1: *An illustrative example for decentralized SGD with (a) synchronous and (b) asynchronous updates in an arbitrary network topology with 4 workers. DSGD suffers from stragglers in (a), while workers in (b) suffer from staleness of the stragglers' parameters, e.g., worker 4 has updated its local parameter for 3 times at iteration $k = 7$, while worker 2 has only updated once. Since worker 2 may randomly pick worker 4 for averaging as in AD-PSGD Lian et al. (2018), worker 4's parameter update could be significantly dragged down by worker 2's stale information.*

iteration $k$, only a subset of $\mathcal{N}(k) \subset \mathcal{N}$ workers participate, and each worker $j$ only waits for a subset of neighbors, i.e., $\mathcal{N}_j(k) \subset \mathcal{N}_j$ to perform local model updates, where we have $\mathcal{N}(k) = \bigcup \mathcal{N}_j(k)$. We denote the cardinality of $\mathcal{N}(k)$ and $\mathcal{N}_j(k)$ as $a(k) := |\mathcal{N}(k)| < N$ and $a_j(k) := |\mathcal{N}_j(k)| < |\mathcal{N}_j|$, respectively. On one hand, these $a(k)$ or $a_j(k)$ workers are the fastest computational workers, which addresses the straggler issues in synchronous updates. On the other hand, each worker does not need to perform local update immediately using stale information or communicate with all neighbors, which addresses the stateless and heavy-communication issues in fully asynchronous updates.

As a result, each worker $j \in \mathcal{N}(k)$ first locally updates its parameter at iteration $k$ and then follows the consensus update only with workers from $\mathcal{N}_j(k)$, i.e.,

$$
\begin{aligned}
\tilde{\mathbf{w}}_j(k) &= \mathbf{w}_j(k-1) - \eta g_j(\mathbf{w}_j(k-1)), \\
\mathbf{w}_j(k) &= \sum_{i \in \mathcal{N}_j(k)} \tilde{\mathbf{w}}_i(k) P_{i,j}(k),
\end{aligned}
\tag{4}
$$

where $\mathbf{P}(k) = (P_{i,j}(k))$ is the time-varying stochastic consensus matrix at iteration $k$, and $P_{i,j}(k) = 0$ if $i \in \mathcal{N} \setminus \mathcal{N}_j(k)$. We denote the stochastic gradient matrix as $\mathbf{G}(k) = [g_1(\mathbf{w}_1(k)), \ldots, g_N(\mathbf{w}_N(k))] \in \mathbb{R}^{d \times N}$ and the learning rate as $\boldsymbol{\eta} = \mathrm{diag}[\eta, \ldots, \eta] \in \mathbb{R}^{N \times N}$. From the above definition, we have $g_j(\mathbf{w}_j(k)) = \mathbf{0}^{d \times 1}$ for $j \notin \mathcal{N}(k)$. Therefore, the compact form of (4) is

$$
\mathbf{W}(k) = [\mathbf{W}(k-1) - \boldsymbol{\eta}\mathbf{G}(k-1)]\mathbf{P}(k).
\tag{5}
$$

The value of $a(k)$ and hence the consensus matrix $\mathbf{P}(k)$ in our AAU model is not static but dynamically changes over iterations by nature. We summarize the above procedure in Algorithm 1, and call the resulting *decentralized SGD with adaptive asynchronous updates* algorithm as `DSGD-AAU`.

There are two open questions for `DSGD-AAU`. First, can `DSGD-AAU` still achieve a near-optimal convergence rate as in state-of-the-art decentralized method with synchronous updates Lian et al. (2017) given that the analysis of asynchronous algorithm is already quite challenging? Second, how to determine $\mathcal{N}(k)$ (line 2 in Algorithm 1), and hence the number of fastest $a(k)$ workers at iteration $k$ to not only maintain the aforementioned best-of-both-world properties, but minimize the global training (convergence) time over the network? Note that at each iteration $k$ in Algorithm 1, the key is to determine the set of fastest workers, i.e., $\mathcal{N}(k) \in \mathcal{N}$, which have completed the local gradient computations, and only workers in set $\mathcal{N}(k)$ actively conduct gossip-average with neighbors while workers in set $\mathcal{N} \setminus \mathcal{N}(k)$ are still computing their local gradients. The iteration $k$ begins at finding the set $\mathcal{N}(k)$ and ends when $\mathcal{N}(k)$ is determined with workers inside it having completed the local gradient computations and the gossip-average operations. In the following, we provide an affirmative answer to the first question in Section 4, and then present a practical algorithm to determine $\mathcal{N}(k)$ (See Algorithm 2 for a fully decentralized implementation in Section 5).

---

**Algorithm 1** Decentralized SGD with Adaptive Asynchronous Updates (`DSGD-AAU`)

---

**Input:** Topology $\mathcal{G} = (\mathcal{N}, \mathcal{E})$, iteration numbers $K$, and initialized $\{\mathbf{w}_j(0)\}_{j=1}^N$.

 1: **for** $k = 0, 1, ..., K-1$ **do**
 2:     Determine the subset of workers $\mathcal{N}(k)$;
 3:     **for** $j = 1, 2, \ldots, N$ and $j \in \mathcal{N}(k)$ **do**
 4:         Update $\tilde{\mathbf{w}}_j(k) = \mathbf{w}_j(k) - \eta g_j(\mathbf{w}_j(k), \mathcal{C}_j(k))$;
 5:         Update $\mathbf{w}_j(k+1) = \sum_{i \in \mathcal{N}_j(k)} \tilde{\mathbf{w}}_i(k) P_{i,j}(k)$;
 6:     **end for**
 7:     Update $\mathbf{w}_j(k+1) = \mathbf{w}_j(k)$ if $j \notin \mathcal{N}(k)$.
 8: **end for**

---

## 4 Convergence Analysis

In this section, we analyze the convergence of `DSGD-AAU`. We show that a linear speedup for convergence (i.e., convergence performance increases linearly with respect to the number of workers) is achievable by `DSGD-AAU`. We first state our assumptions as follows.

### 4.1 Assumptions

**Assumption 4.1** (Non-Negative Metropolis Weight Rule)**.** The Metropolis weights on a time-varying graph $(\mathcal{N}, \mathcal{E}_k)$ with consensus matrix $\mathbf{P}(k) = (P_{i,j}(k))$ at iteration $k$ satisfy

$$\begin{cases} P_{i,i}(k) = 1 - \sum_{j \in \mathcal{N}_i(k)} P_{i,j}(k), \\ P_{i,j}(k) = \frac{1}{1 + \max\{p_i(k), p_j(k)\}}, & \text{if } j \in \mathcal{N}_i(k), \\ P_{i,j}(k) = 0, & \text{otherwise,} \end{cases} \tag{6}$$

where $p_i(k)$ is the number of active neighbors that worker $i$ needs to wait at iteration $k$. Given the Metropolis weights, $\mathbf{P}(k)$ are doubly stochastic, i.e., $\sum_{j=1}^N P_{i,j}(k) = \sum_{i=1}^N P_{i,j}(k) = 1, \forall i \in [N], j \in [N], k$.

**Assumption 4.2** (Bounded Connectivity Time)**.** There exists an integer $B \geq 1$ such that the graph $\mathcal{G} = (\mathcal{N}, \mathcal{E}_{kB} \cup \mathcal{E}_{kB+1} \cup \ldots \cup \mathcal{E}_{(k+1)B-1}), \forall k$ is strongly-connected.

**Assumption 4.3** (L-Lipschitz Continuous Gradient)**.** There exists a constant $L > 0$, such that $\|\nabla F_j(\mathbf{w}) - \nabla F_j(\mathbf{w}')\| \leq L\|\mathbf{w} - \mathbf{w}'\|, \ \forall j \in [N], \forall \mathbf{w}, \mathbf{w}' \in \mathbb{R}^d$.

**Assumption 4.4** (Unbiased Local Gradient Estimator)**.** The local gradient estimator is unbiased, i.e., $\mathbb{E}[g_j(\mathbf{w})] = \nabla F_j(\mathbf{w}), \forall j \in [N], \mathbf{w} \in \mathbb{R}^d$ with the expectation taking over the local dataset samples.

**Assumption 4.5** (Bounded Variance)**.** There exist two constants $\sigma_L > 0$ and $\varsigma > 0$ such that the variance of each local gradient estimator is bounded by $\mathbb{E}[\|g_j(\mathbf{w}) - \nabla F_j(\mathbf{w})\|^2] \leq \sigma_L^2, \forall j \in [N], \mathbf{w} \in \mathbb{R}^d$, and the global variability of the local gradient of the loss function is bounded by $\|\nabla F_j(\mathbf{w}) - \nabla F(\mathbf{w})\|^2 \leq \varsigma^2, \forall j \in [N], \mathbf{w} \in \mathbb{R}^d$.

Assumptions 4.1-4.5 are standard in the literature. Assumptions 4.1 and 4.2 are common in dynamic networks with finite number of nodes Xiao et al. (2006); Tang et al. (2018a). Specifically, Assumption 4.1 guarantees the product of consensus matrices $\mathbf{P}(1) \cdots \mathbf{P}(k)$ being a doubly stochastic matrix, while Assumption 4.2 guarantees the product matrix is a strictly positive matrix for large $k$. We define $\mathbf{\Phi}_{k:s}$ as the product of consensus matrices from $\mathbf{P}(s)$ to $\mathbf{P}(k)$, i.e., $\mathbf{\Phi}_{k:s} \triangleq \mathbf{P}(s)\mathbf{P}(s+1) \cdots \mathbf{P}(k)$. Under Assumption 2, Nedić et al. (2018) proved that for any integer $\ell, n, \mathbf{\Phi}_{(\ell+n)B-1:\ell B}$ is a strictly positive matrix. In fact, every entry of it is at least $\beta^{nB}$, where $\beta$ is the smallest positive value of all consensus matrices, i.e., $\beta = \arg\min_{i,j,k} P_{i,j}(k)$ with $P_{i,j}(k) > 0, \forall i, j, k$. Due to the strongly-connected graph search procedure, Assumption 4.2 naturally holds for `DSGD-AAU` and $B \leq N-1$. Assumption 4.3 is widely used in convergence results of gradient methods, e.g., Nedic & Ozdaglar (2009); Bubeck et al. (2015); Bottou et al. (2018); Nedić et al. (2018). Assumptions 4.4-4.5 are also standard Kairouz et al. (2019); Tang et al. (2020); Yang et al. (2021). We use a universal bound $\varsigma$ to quantify the heterogeneity of the non-i.i.d. datasets among different workers. If $\varsigma = 0$, the datasets of all workers are i.i.d. It is worth noting that we do *not* require a bounded gradient assumption, which is often used in distributed optimization analysis Nedic & Ozdaglar (2009); Nedić & Olshevsky (2016).

## 4.2 Convergence Analysis for `DSGD-AAU`

We now present the main result for `DSGD-AAU`.

**Theorem 4.6.** *Choose constant learning rates $\eta \leq \min \left( \sqrt{\frac{(1-q)^2}{30C^2L^2N} + \frac{9N^4}{16}} - \frac{3N^2}{4}, 1/L \right)$, where $C := \frac{1+\beta^{-NB}}{1-\beta^{NB}}$ and $q := (1 - \beta^{NB})^{1/NB}$. Under Assumptions 4.1-4.5, the sequence of parameter $\{\mathbf{w}_j(k), \forall j \in [N]\}$ generated by* `DSGD-AAU` *satisfies*

$$\frac{1}{K} \sum_{k=0}^{K-1} \mathbb{E} \left\| \nabla F \left( \frac{\sum_{j=1}^{N} \mathbf{w}_j(k)}{N} \right) \right\|^2 \leq \frac{6 \left( F(\bar{\mathbf{w}}_0) - F(\mathbf{w}^*) \right)}{\eta K} + \frac{(9L+2)\eta}{3N} \sigma_L^2 + \frac{2\eta}{N} \varsigma^2, \tag{7}$$

*where $\bar{\mathbf{w}}_0 := \frac{\sum_{j=1}^{N} \mathbf{w}_j(0)}{N}$, $\mathbf{w}^*$ is the optimal parameter, and the expectation is taken over the local dataset samples among workers.*

*Remark* 4.7. The right hand side of (7) consists of two parts: (i) a vanishing term $6(F(\bar{\mathbf{w}}(0)) - F(\mathbf{w}^*))/\eta K$ that goes to zero as $K$ increases; and (ii) a constant term $\frac{(9L+2)\eta}{3N}\sigma_L^2 + \frac{2\eta}{N}\varsigma^2$ depending on the problem instance parameters and is independent of $K$. The vanishing term's decay rate matches that of the typical DSGD methods, and the constant term converges to zero independent of $K$ as $N$ increases.

The key to achieve the bound in (7) boils down to bound the gap between individual parameter $\mathbf{w}_j(k)$ and the average $\bar{\mathbf{w}}(k) := \sum_j \mathbf{w}_j(k)/N$, which is significantly affected by asynchronous updates in (4). In addition, existing state-of-the-art asynchronous method AD-PSGD Lian et al. (2018) assumes a predefined distribution of update frequency for all workers and the global dataset should be uniformly distributed to each worker, while neither is required by our `DSGD-AAU`. These make their approaches not directly applied to ours and hence necessitate different proof techniques. In particular, to bound $\mathbb{E}\|\mathbf{w}_j(k) - \bar{\mathbf{w}}(k)\|^2$, AD-PSGD assumes the spectral gap and bounded staleness of each worker's local update, while we only need to leverage the minimum non-negative element of each consensus matrices (Assumption 4.2). Further differentiating our proof is that we upper bound staleness periods in `DSGD-AAU` by $N - 1$, while existing works often assume a fixed bounded stateless period, which may not hold true in practice.

With Theorem 4.6, we immediately have the following convergence rate for `DSGD-AAU` with a proper choice of learning rate:

**Corollary 4.8.** *Let $\eta = \sqrt{N/K}$. The convergence rate of* `DSGD-AAU` *is $\mathcal{O}\left(\frac{1}{\sqrt{NK}}\right)$ when $K \geq 30C^2L^2N^2$.*

*Remark* 4.9. Our consensus-based decentralized optimization method with adaptive asynchronous updates still achieves a linear speedup $\mathcal{O}\left(\frac{1}{\sqrt{NK}}\right)$ with proper learning rates as shown in Corollary 4.8. Although many works have achieved this convergence rate asymptotically, e.g., Lian et al. (2017) is the first to provide a theoretical analysis of decentralized SGD with a convergence rate of $\mathcal{O}(\frac{1}{\sqrt{NK}} + \frac{1}{K})$, these results are only for consensus-based method with synchronous updates. To the best of our knowledge, we are the first to reveal this result for fully decentralized learning with adaptive asynchronous updates, which is non-trivial due to the large-scale distributed and heterogeneous nature of training data across workers.

An interesting observation is that the value of $C > 1$ is monotonically increasing as $B$ increases. Therefore, the required iteration number $K$ will become larger if the bounded-connectivity time $B$ increases. Hence, reducing the value of $B$ can lower the required number of iterations $K$. This key observation motivates our design of `DSGD-AAU` in Section 5, which aims to jointly dynamically select the active workers per round and minimize the connectivity time $B$. This resolves the staleness issue in conventional asynchronous algorithms Luo et al. (2020); Assran & Rabbat (2020); Wang et al. (2022).

## 5 Realization of `DSGD-AAU`

In this section, we propose a practical algorithm to dynamically determine $\mathcal{N}(k)$ in Algorithm 1 (line 2) in a fully decentralized manner so that we can implement `DSGD-AAU` in real-world systems. For abuse of name, we still call the resulting algorithm `DSGD-AAU`, which is implemented and numerically evaluated in Section 6.

---

**Algorithm 2** Implementation of `DSGD-AAU`: logical view

---

**Input:** $\mathcal{G} = (\mathcal{N}, \mathcal{E})$, iteration numbers $K$, initialized $\{\mathbf{w}_j(0)\}_{j=1}^N$, and empty set $\mathcal{P}, \mathcal{V}$.

1: **for** $k = 0, 1, ..., K-1$ **do**
2:     Randomly select a worker $j_k$ from the set $\mathcal{N}$;
3:     Randomly sample a batch $\mathcal{C}_{j_k}(k)$ from the local data set $\mathcal{D}_{j_k}$ and compute gradient $g_{j_k}(\mathbf{w}_{j_k}(k), \mathcal{C}_{j_k}(k))$;
4:     Update $\tilde{\mathbf{w}}_{j_k}(k) = \mathbf{w}_{j_k}(k) - \eta g_{j_k}(\mathbf{w}_{j_k}(k), \mathcal{C}_{j_k}(k))$;
5:     Run $(\mathcal{N}_{j_k}(k), \mathcal{P}, \mathcal{V}) \leftarrow$ `Pathsearch`$(\mathcal{P}, \mathcal{V}, \mathcal{G}, j_k)$;
6:     Update $\mathbf{w}_{j_k}(k+1) = \sum_{i \in \mathcal{N}_{j_k}(k)} \tilde{\mathbf{w}}_i(k) P_{i,j_k}(k)$;
7:     **for** Any other worker $i_k$ finished the local gradient update $\tilde{\mathbf{w}}_{i_k}(k)$ during `Pathsearch` **do**
8:         Update $\mathbf{w}_{i_k}(k+1) = \sum_{i \in \mathcal{N}_{i_k}(k)} \tilde{\mathbf{w}}_i(k) P_{i,i_k}(k)$;
9:     **end for**
10:     Reset $\mathcal{P} = \phi$ and $\mathcal{V} = \phi$ if graph $\mathcal{G}' = (\mathcal{V}, \mathcal{P})$ is strongly-connected with $\mathcal{V} = \mathcal{N}$.
11: **end for**

---

**Intuition.** Observing that for the parameters of different workers in the decentralized optimization framework to converge, each worker needs to frequently participate in the parameter update, such that every worker's information diffuses to all other workers in the system Nedic & Ozdaglar (2009); Nedić & Olshevsky (2016). This is equivalent to the fact that any two arbitrary workers exchange parameters directly or indirectly through a path as the graph dynamically changes over time. The frequency for establishing such paths is mainly affected by the computation speed of stragglers. In synchronous updates, such paths are established once per "synchronous iteration", determined by the slowest workers (see Figure 1a); while in fully asynchronous updates, such a path for any worker is established immediately in each "asynchronous iteration" when it completes its local updates (see Figure 1b), suffering from stale information as aforementioned. Instead, we propose `DSGD-AAU` to dynamically determine the subset of neighboring workers $\mathcal{N}_j(k)$ for each worker $j \in [N]$ in each iteration $k$ to establish such paths for parameter updates.

**Realization.** In particular, for a given communication graph $\mathcal{G} = \{\mathcal{N}, \mathcal{E}\}$, `DSGD-AAU` establishes the same strongly-connected graph *at each worker's side in a fully decentralized manner* such that there exists at least one path for arbitrary two nodes/workers $i, j \in [N]$. We denote the set that contains all edges of such a strongly-connected graph as $\mathcal{P}$ and the set for all corresponding vertices as $\mathcal{V}$, which is essentially equal to $\mathcal{N}$. Our key insight is that all edges in $\mathcal{P}$ have been visited by `DSGD-AAU` at least once, i.e., all nodes in $\mathcal{V}$ share information with each other. Specifically, `DSGD-AAU` establishes sub-graphs $\mathcal{G}' := \{\mathcal{P}, \mathcal{V}\}$ to dynamically determine the number of workers involved in parameter updates at *each "asynchronous iteration"* in a fully decentralized manner, which requires each worker to store a local copy of $\mathcal{P}$ and $\mathcal{V}$, i.e., $\mathcal{P}_j, \mathcal{V}_j, \forall j \in [N]$, and then to reach a consensus on $\mathcal{P}$ and $\mathcal{V}$, i.e., $\mathcal{P} = \mathcal{P}_j, \mathcal{V} = \mathcal{V}_j, \forall j$ whenever any change occurs for arbitrary $\mathcal{P}_j$ and $\mathcal{V}_j$. When a new edge $(i, j)$ is established satisfying $(i, j) \notin \mathcal{P}_j, \mathcal{P}_i$, and either $j \notin \mathcal{V}_i$ or $i \notin \mathcal{V}_j$, workers $i$ and $j$ broadcast the information $(i, j)$ to the entire network such that *each worker will update her own* $\mathcal{P}_{j'}, \forall j' \in [N]$ *locally.* Specifically, for any worker $j'$, whenever she receives any new edges and vertices from her neighbors in $\mathcal{N}_{j'}$, worker $j'$ broadcasts this new information to her neighbors. This information sharing process continues until no new information comes. Eventually, each worker's local set $\mathcal{P}_j, \mathcal{V}_j$ reaches a consensus, i.e., $\mathcal{P} = \mathcal{P}_j, \mathcal{V} = \mathcal{V}_j, \forall j$. This procedure also avoids deadlock issues. Once the last path $(i, j)$ is established, $\mathcal{G}' := \{\mathcal{P}, \mathcal{V}\}$ is strongly connected with $\mathcal{V} = \mathcal{N}$. A detailed description on how to find new edges and vertices in $\mathcal{P}$ and $\mathcal{V}$ (i.e., the `Pathsearch`) is provided in the appendices.

In the following, we will directly use $\mathcal{G}' = (\mathcal{P}, \mathcal{V})$ to describe `DSGD-AAU` for simplicity, under the assumption that each worker has reached a consensus on $\mathcal{P}_j$ and $\mathcal{V}_j, \forall j \in [N]$. Our `DSGD-AAU` algorithm (summarized in Algorithm 2) can be described in the following. We use a virtual counter $k$ to denote the iteration counter – *every new edge* is established no matter on which workers will increase $k$ by 1. Let $j_k$ be the first worker which completes the local gradient computation in the $k$-th iteration. At each iteration $k$, the worker $j_k$ performs a local SGD step (lines 2-4) followed by one step of the graph searching procedure to establish a new edge in set $\mathcal{P}$ (line 5). The next operation in current iteration is the gossip-average of the local parameters of

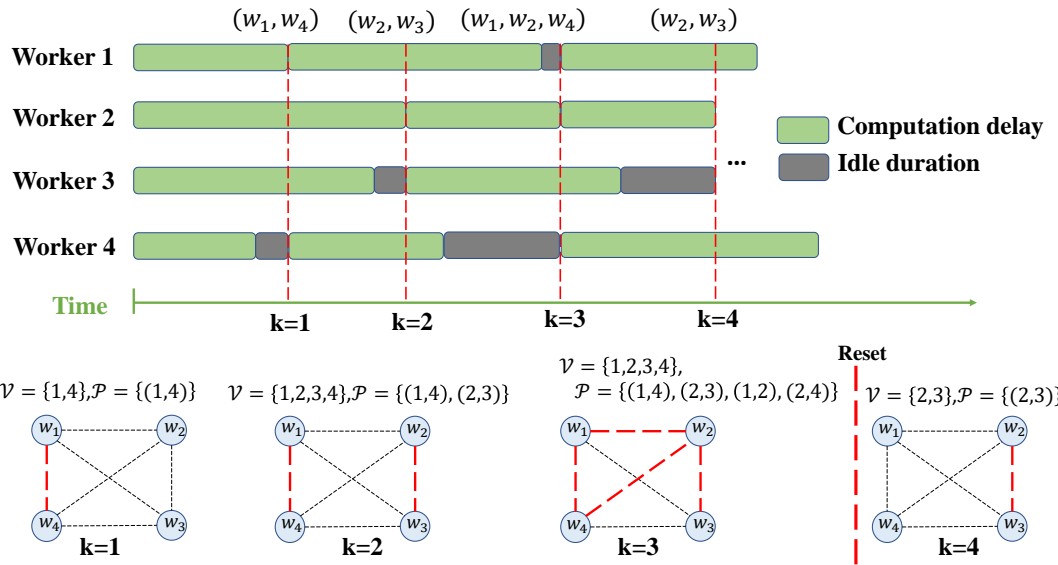

Figure 2: *An illustrative example of* `DSGD-AAU` *for 4 heterogeneous workers with a fully-connected network topology.*

workers $j_k$ in the set $\mathcal{N}_{j_k}(k)$ (line 6). Note that if any other worker $i_k$ finishes local gradient update during the `Pathsearch` of the fast worker $j_k$, it will also run `Pathsearch` independently until $\mathcal{P}$ and $\mathcal{V}$ get updated. Hence, worker $i_k$ also conduct gossip-average of the local parameters in the set $\mathcal{N}_{i_k}(k)$ (lines 7-9). When a strongly-connected graph $\mathcal{G}' = (\mathcal{P}, \mathcal{V})$ is established, reset $\mathcal{P}$ and $\mathcal{V}$ as empty sets (line 10).

`Pathsearch.` At each iteration $k$, we denote the worker which first completes the local parameter update as $j_k$. Worker $j_k$ keeps idle until any one of the following two situations occurs. The first case is that one of worker $j_k$'s neighbor $i \in \mathcal{N}_{j_k}$ completes the local parameter update. If edge $(j_k, i) \in \mathcal{E} \cap (j_k, i) \notin \mathcal{P}$ and worker $j_k \notin \mathcal{V} \bigcup i \notin \mathcal{V}$, then store edge $(j_k, i)$ into $\mathcal{P}$ and nodes $\{i, j_k\}$ into $\mathcal{V}$. Once this is achieved, all workers move into the $(k+1)$-th iteration. The other case is that any other new edge $(i_1, j_1)$ between workers $i_1$ and $j_1$ other than $j_k$ is established such that edge $(i_1, j_1) \in \mathcal{E} \cap (i_1, j_1) \notin \mathcal{P}$ and node $j_1 \notin \mathcal{V} \bigcup i_1 \notin \mathcal{V}$. Then all workers move to the $(k+1)$-th iteration. Intuitively, for both cases we guarantee that there is at least one new worker (with new updated parameters) involved in each iteration, no matter the new worker is a neighbor or for of worker $j_k$. This process is repeated for each iteration, which dynamically determines the subset of the fastest workers $\mathcal{N}(k)$. A detailed description of `Pathsearch` is provided in Algorithm 3 in Section B in the appendices. An illustrative example of `DSGD-AAU` for 4 heterogeneous workers with a fully-connected network topology is presented in Figure 2. At the 1st iteration ($k = 1$), worker 4 first completes the local computation, it waits until the neighbor worker 1 finishes the local computation. Then, workers 4 and 1 exchange the updates and `DSGD-AAU` stores nodes $1, 4$ into $\mathcal{V}$, and edge $(1, 4)$ into $\mathcal{P}$. This terminates the 1st iteration. At the 3rd iteration ($k = 3$), worker 4 first completes the local computation, it waits until worker 1 completes the local computation. However, since nodes $1, 4$ have been stored in $\mathcal{V}$ and edge $(1, 4) \in \mathcal{P}$, worker 4 keeps waiting until worker 2 completes the local computation. Workers $1, 2, 4$ exchange updated information, and `DSGD-AAU` stores new worker 2 in $\mathcal{V}$ and $\mathcal{P} = \{(1, 4), (2, 3), (1, 2), (2, 4)\}$, which ends the 3rd iteration. Since there exists a path for any arbitrary two workers in $\mathcal{G}' := \{\mathcal{P}, \mathcal{V}\}$ with $\mathcal{V} = \mathcal{N}$, `DSGD-AAU` resets $\mathcal{V}$ and $\mathcal{P}$ as empty sets and move to the next iteration.

*Remark* 5.1. The procedures in Algorithm 2 can be executed by each worker independently, and thus enables a fully decentralized implementation. Need to mention that ID sharing requires a much lower communication overhead compared with parameters sharing among neighbors. Hence no extra overhead is introduced. The overall communication overhead for each worker is upper-bounded by $\mathcal{O}(2NB)$ with $B$ being the maximum number of iterations for searching a strongly-connected graph $\mathcal{G}' := \{\mathcal{P}, \mathcal{V}\}$, which is smaller than $N - 1$. In addition, `DSGD-AAU` needs a memory size no more than $\mathcal{O}(N)$ to store a local copy of $\mathcal{P}$ and $\mathcal{V}$ as aforementioned. One of our key contributions is the Pathsearch procedure, which is essentially used to determine when a logical iteration ends. The Pathsearch procedure ends at each iteration only

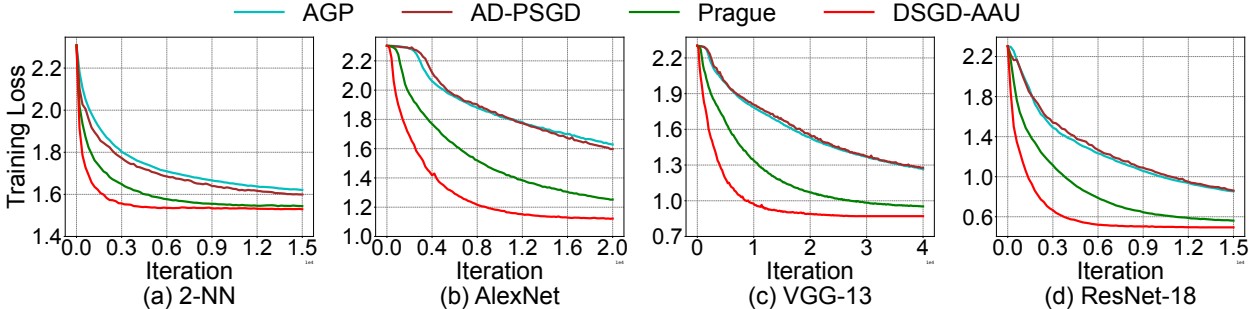

Figure 3: Training loss w.r.t. *iteration* for different models on non-i.i.d. CIFAR-10 with 128 workers.

|  | AGP | AD-PSGD | Prague | DSGD-AAU |
|---|---|---|---|---|
| 2-NN | $43.87 \pm 0.15$ | $43.55 \pm 0.13$ | $44.51 \pm 0.19$ | $\mathbf{45.43} \pm 0.16$ |
| AlexNet | $52.85 \pm 0.11$ | $49.54 \pm 0.16$ | $56.32 \pm 0.17$ | $\mathbf{58.61} \pm 0.12$ |
| VGG-13 | $59.41 \pm 0.15$ | $56.30 \pm 0.19$ | $63.52 \pm 0.17$ | $\mathbf{67.14} \pm 0.14$ |
| ResNet-18 | $76.25 \pm 0.11$ | $73.72 \pm 0.12$ | $77.36 \pm 0.13$ | $\mathbf{79.80} \pm 0.18$ |

Table 1: Test accuracy for different models on non-i.i.d. CIFAR-10 with 128 workers.

when a new link is established in $\mathcal{P}$. This definition of an iteration in DSGD-AAU is different from the one in AD-PSGD where a single gradient update on whichever worker is counted as one iteration. In DSGD-AAU, an iteration involves a more complex process that includes a local SGD step, a graph searching procedure, and a gossip-average operation. Moreover, since Pathsearch guarantees at each iteration, a new worker must be involved in the parameter exchange, and all workers will participate at least once every $N - 1$ iterations. This is also a key property of our proposed DSGD-AAU.

# 6 Numerical Results

We conduct extensive experiments to validate our model and theoretical results. We implement DSGD-AAU in PyTorch Paszke et al. (2017) on Python 3 with three NVIDIA RTX A6000 GPUs using the Network File System (NFS) and MPI backends, where files are shared among workers through NFS and communications between workers are via MPI. We relegate some experimental details, parameter settings and a comprehensive set of results to Section D in the appendices.

**Baseline.** We compare DSGD-AAU with three state-of-the-art methods: AD-PSGD Lian et al. (2018), Prague Luo et al. (2020) and AGP Assran & Rabbat (2020), where workers are randomly selected for parameter averaging in AD-PSGD and Prague. Hence both suffer from stragglers in the optimization problem since they ignore slower workers.

**Dataset and Model.** We consider four representative models: a fully-connected neural network with 2 hidden layers (2-NN), AlexNet Krizhevsky et al. (2012), VGG-13 Simonyan & Zisserman (2015) and ResNet-18 He et al. (2016) for classification tasks using CIFAR-10 Krizhevsky et al. (2009), MNIST LeCun et al. (1998), and Tiny-ImageNet Russakovsky et al. (2015) datasets, in particular focusing on non-i.i.d. versions. For example, we use the idea in McMahan et al. (2017) to obtain non-i.i.d. CIFAR-10 (We provide a detailed discussion on how to generate non-i.i.d. dataset in Section D "Additional Experimental Results".). We also investigate the task of next-character prediction on the dataset of *The Complete Works of William Shakespeare* McMahan et al. (2017) (Shakespeare) with an LSTM language model Kim et al. (2016). We consider a network with 32, 64, 128, 256 workers and randomly generate a connected graph for evaluation.

In real systems, there are several reasons leading to the existing of stragglers, e.g., heterogeneous hardware, hardware failure, imbalanced data distributions among tasks and different OS effects, resource contention, etc. Since we run our experiments of different number of workers in one sever as in many other works, e.g.,

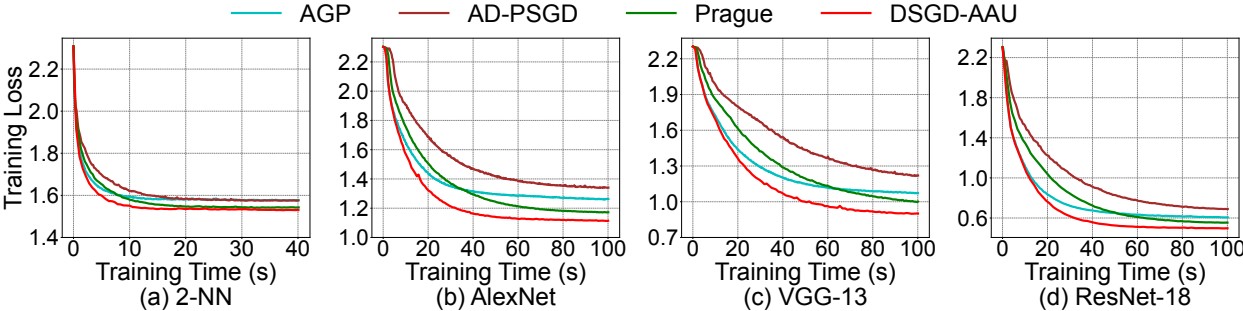

Figure 4: Training loss w.r.t. *time* for different models on non-i.i.d. CIFAR-10 with 128 workers.

Li et al. (Li et al., 2020a;b) and motivated by the idea of AD-PSGD (Lian et al., 2018) and Cipar et al. (Cipar et al., 2013), we randomly select workers as stragglers in each iteration. If a worker is selected to be a straggler, then it will sleep for some time in the iteration (e.g., the sleep time could be 6x of the average one local computation time). We also investigate the impact of the sleep time and the percentage of stragglers in our ablation study, see Section D. Such techniques have been widely used in the literature such as AD-PSGD (Lian et al., 2018), Prague (Luo et al., 2020) and Li et al. (Li et al., 2020a;b).

The loss function is the cross-entropy one. The learning rate is perhaps the most critical hyperparameter in distributed ML optimization problems. A proper value of the learning rate is important; however, it is in general hard to find the optimal value. With the consideration of adaptive asynchronous updates, it is reasonable to adaptively set the learning rate to speed up the convergence process in experiments, as in Chen et al. (2016); Lian et al. (2018); Xu et al. (2020); Assran & Rabbat (2020). Hence, we choose $\eta(k) = \eta_0 \cdot \delta^k$, where $\eta_0 = 0.1$ and $\delta = 0.95$. Finally, batch size is another important hyperparameter, which is limited by the memory and computational resources available at each worker, or determined by generalization performance of final model Hoffer et al. (2017). We test the impact of batch size using these datasets and find that 128 is a proper value. An ablation study on these hyperparameters (e.g. straggler rate) is given in the appendices. For ease of readability, in this section, we only present the numerical results on non-i.i.d. partitoned CIFAR-10, and relegate the corresponding results on other datasets to Section D in the appendices.

**Convergence.** Figure 3 shows the training loss for all algorithms w.r.t. iterations, which are evaluated for different models on non-i.i.d. CIFAR-10 dataset with 128 workers. The corresponding test accuracy is presented in Table 1. We observe that `DSGD-AAU` significantly outperforms AD-PSGD and AGP, and performs better than Prague when the total number of iterations is large enough. Its advantage is especially pronounced with a limited number of iterations or a limited training time (see Figure 4), e.g., within 6,000 iterations, `DSGD-AAU` achieves much smaller loss (resp. higher test accuracy) than Prague for ResNet-18. Finally, note that the test accuracy for non-i.i.d. CIFAR-10 in Table 1 is relatively lower compared to that for i.i.d. dataset, a phenomenon widely observed in the literature Yang et al. (2021); Zhao et al. (2018); Yeganeh et al. (2020); Wang et al. (2020a); Hsieh et al. (2020).

Similar observations can be made with 32, 64, 256 workers across other datasets and models (see the appendices). In particular, we summarize the test accuracy for all algorithms for ResNet-18 on non-i.i.d. CIFAR-10 trained for 50 seconds with different number of workers in Table 2. `DSGD-AAU` also achieves a higher test accuracy compared to AD-PSGD, Prague and AGP.

**Speedup and Communication.** The convergence in wall-clock time of ResNet-18 on non-i.i.d. CIFAR-10 dataset with 128 workers is presented in Figure 4. Again, we observe that `DSGD-AAU` converges faster, and importantly, `DSGD-AAU` achieves a higher accuracy for a given training time. The speedup of different algorithms for ResNet-18 on non-i.i.d. CIFAR-10 with respect to different number of workers is presented in Figure 5(a). We compute the speedup of different algorithms with respect to the DSGD with full worker updates when 65% accuracy is achieved for ResNet-18 model[2]. `DSGD-AAU` consistently outperforms baseline

---

[2]In Figure 5, we report the results using ResNet-18 on non-i.i.d. CIFAR-10, for which the highest test accuracy that "DSGD with full worker updates" can achieve is a little above 65% (note that we consider the non-i.i.d. settings). Since "DSGD with

| # | AGP | AD-PSGD | Prague | DSGD-AAU |
|---|-----|---------|--------|----------|
| 32 | $67.19\% \pm 0.18\%$ | $55.67\% \pm 0.35\%$ | $60.22\% \pm 0.24\%$ | **71.56%** $\pm 0.32\%$ |
| 64 | $73.01\% \pm 0.11\%$ | $63.09\% \pm 0.18\%$ | $69.65\% \pm 0.24\%$ | **77.34%** $\pm 0.23\%$ |
| 128 | $75.18\% \pm 0.13\%$ | $68.64\% \pm 0.15\%$ | $74.72\% \pm 0.12\%$ | **78.58%** $\pm 0.17\%$ |
| 256 | $75.32\% \pm 0.14\%$ | $73.50\% \pm 0.12\%$ | $77.14\% \pm 0.18\%$ | **78.76%** $\pm 0.15\%$ |

Table 2: Test accuracy for ResNet-18 on non-i.i.d. CIFAR-10 trained for 50 seconds with different number of workers.

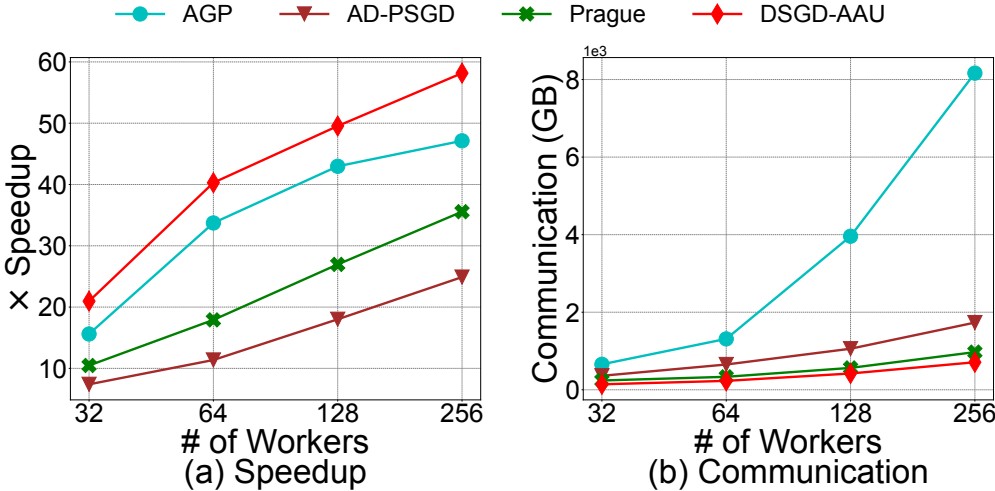

Figure 5: Speedup and communication for ResNet-18 on non-i.i.d. CIFAR-10 with different number of workers.

methods and achieves the best speedup at no cost of additional communication (i.e., network transmission for training) in the system as shown in Figure 5(b). Similar observations can be made for other models with different datasets (see the appendices), i.e., DSGD-AAU is robust across different models and datasets to achieve the fastest convergence time and the best speedup while maintaining comparable or even better communication efficiency. Need to mention that our convergence time in the numerical evaluation considers the total time of the whole process, which includes both computation and communication time. since we mainly focus on computational stragglers in this paper, the communication time are dominated by computational time in our considered setting. A more detailed discussion can be found in Section C.4 in the appendices.

# 7 Related Work

Our algorithm belongs to the class of consensus/gossip algorithms, or fully decentralized/peer-to-peer learning algorithms Kairouz et al. (2019); Yang et al. (2019); Imteaj et al. (2021; 2022). There is a rich literature of consensus algorithms based on the theory of Tsitsiklis et al. (1986) in various fields including Bertsekas & Tsitsiklis (1989); Kempe et al. (2003); Xiao & Boyd (2004); Boyd et al. (2006); Nedic & Ozdaglar (2009); Johansson et al. (2010); Ram et al. (2010); Boyd et al. (2011); Duchi et al. (2011); Colin et al. (2016); Scaman et al. (2017; 2018); Tang et al. (2018a;b); Li et al. (2019); Neglia et al. (2020). However, most results are for full worker participation with synchronous updates in each iteration, which can be communication expensive. Efficient control of adaptive asynchronous updates in consensus-based distributed optimization has received little attention, which arise in problems of practical interests. The few existing works address this issue through a stale-synchronous model Chen et al. (2016); Ho et al. (2013) with no convergence guarantee, and the number of fastest neighbor workers is often configured manually through preliminary experiments before the start of the actual training process.

---

full worker updates" is the baseline to compute the speedup for all algorihtms we compare, including AGP, AD-PSGD, Prague and our DSGD-AAU, we choose the 65% as the target test accuracy.

The asynchronous SGD Agarwal & Duchi (2011) breaks the synchronization in synchronous SGD by allowing workers to use stale weights to compute gradients, which significantly reduces the idel time for synchronization. Lian et al. (2018) proposed the first asynchronous decentralized parallel SGD algorithm AD-PSGD which achieves linear speedup with the number of workers. The limitation of AD-PSGD is that the network topology needs be bipartite to avoid deadlock issue. Luo et al. (2020) proposed Prague based on a slight modification of AD-PSGD, which resolves the deadlock issue and further reduces the convergence times. Assran & Rabbat (2020) proposed asynchronous gradient push (AGP) to deal with directed graph with non-doubly stochastic consensus matrix under the assumption that the loss function is convex. A similar gradient-push technique was used for decentralized online learning in Jiang et al. (2021). For non-convex stochastic optimization on directed graphs, Kungurtsev et al. (2021) proposed an asynchronous algorithm which combines stochastic gradients with tracking in an asynchronous push-sum framework and obtains the standard sublinear convergence rate. Nadiradze et al. (2021) proposed an asynchronous decentralized SGD with quantized local updates to reduce the communication overhead. Wang et al. (2022) proposed a predicting Clipping asynchronous SGD where the predicting step leverages the gradient prediction using Taylor expansion to reduce the staleness of the outdated weights while the clipping step selectively drops the outdated weights to alleviate their negative effects. More related work can be found in Assran et al. (2020). A more detailed review of fully decentralized learning algorithms as well as the All-Reduce Li et al. (2020a;b), or the parameter-server paradigm Karakus et al. (2017); Charalambides et al. (2020); Glasgow & Wootters (2021), such as the popular federated learning McMahan et al. (2017), is provided in the appendices.

Although we focus on stragglers caused by heterogeneous computation capabilities among workers as in previous work in (Ananthanarayanan et al., 2013; Ho et al., 2013; Lian et al., 2017; 2018; Luo et al., 2020),there is another line of works focusing on another key type of stragglers, i.e., the communication stragglers Wang et al. (2019); Wang & Joshi (2021); Wang et al. (2020b). In particular, the celebrated "MATCHA" algorithm was proposed in Wang et al. (2019) to mitigate the communication stragglers. Note that although MATCHA also has a link-control procedure, it differentiates from our proposed `DSGD-AAU` algorithm fundamentally. Specifically, `DSGD-AAU` adaptively selects the fastest computational workers for each iteration to mitigate the effect of slower computational workers, while MATCHA optimizes runtime by tuning the frequency of inter-node communication, potentially reducing the impact of slower communication workers, which leads `DSGD-AAU` to be an asynchronous algorithm and MACTHA to be a synchronous one. Furthermore, in contrast to MATCHA, our `DSGD-AAU` leverages the bounded-connectivity time to theoretically show the convergence rather than using the conventional spectral radius as in Lian et al. (2017; 2018). These differences highlight that `DSGD-AAU` and MATCHA are designed to address distinct challenges in the field.

## 8 Conclusion

In this paper, we considered to mitigate the effect of stragglers in fully decentralized learning via adaptive asynchronous updates. We proposed `DSGD-AAU` and proved that `DSGD-AAU` achieves a linear speedup for convergence, and dramatically reduces the convergence time compared to conventional methods. Finally, we provided extensive empirical experiments to validate our theoretical results.

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

# A   Additional Related Work

We provide additional discussions on the related work. In particular, we compare our `DSGD-AAU` with two state-of-the-arts AD-PSGD (Lian et al., 2018) and Prague (Luo et al., 2020). Similar to traditional training such as PS and All-Reduce, in one iteration, AD-PSGD computes gradients first, and then performs synchronization; the difference is that it only synchronizes with a randomly selected neighbor, instead of all other workers. However, a conflict will occur in AD-PSGD when two workers simultaneously select one worker for synchronization. In order to keep the atomic property of weight updating, the two operations need to be serialized, which slows down the whole process. To resolve the conflict issue and fasten the synchronization, Prague proposed a partial All-Reduce scheme which randomly generates a group for each node, and the node only communicates with other nodes in the same group. However, the performance of Prague heavily depends on a so-called "Group Generator", i.e., when one worker completes its local update, it would inquire the Group Generator to get its group number, and then it has to wait all members in the corresponding group to complete their local updates before implementing the communication process (i.e., sharing the local updates with the group members). In particular, three methods were proposed in (Luo et al., 2020) including an offline static method, a randomized method and a "smart" method using a heuristic threshold-based rule. However, the method of how to adjust the threshold was not discussed.

Need to mention that there is no specific rule to differentiate stragglers or non-stragglers in AD-PSGD and Prague. As a result, both are still affected by stragglers because the randomly selection (even through a heuristic threshold-based rule) of communicating workers has the chance of taking the stragglers into account, which eventually slow down the training process. In contrast, our `DSGD-AAU` (Algorithm 1 in the main paper) dynamically determines the number of fastest neighbor workers for each worker and differentiate stragglers iteration-by-iteration based on the running time (`Pathsearch` in Algorithm 3), such that it is capable of mitigating the effect from stragglers at each iteration. From `DSGD-AAU` (Algorithm 1 in the main paper), we can anticipate that our algorithm requires fewer iterations for convergence compared with Prague and AD-PSGD because the average number of participating workers in our algorithm is larger than that of the benchmarks. This is verified by the numerical results.

In addition, a similar idea of adaptive asynchronous update has been widely used in DML systems, there exist significant challenges. Previous works mostly focused on the PS or the All-Reduce models, and most of existing algorithms are network topology-constrained and are hard to be generalized and implemented in arbitrary network topologies as our proposed `DSGD-AAU` does. Moreover, most algorithms including in Li et al. (Li et al., 2020a;b) only select the fastest few workers for the parameter averaging process, where a lot of "normal" workers are ignored to drag down the whole update process. Furthermore, another line of works using coding techniques such as (Karakus et al., 2017; Charalambides et al., 2020; Glasgow & Wootters, 2021) aim to recover the stragglers' gradient information by leveraging gradient coding techniques from information theory perspective. However, same data needs to be sent to multiple nodes to enable gradient encoding in these works, while the dataset for each node is disjoint in our model. Hence each node in our model has non-i.i.d. distinct dataset, while cannot be applied in (Karakus et al., 2017; Charalambides et al., 2020; Glasgow & Wootters, 2021).

A more relevant line of research on distributed learning is the All-Reduce (Li et al., 2020a;b), or the PS (Karakus et al., 2017; Charalambides et al., 2020; Glasgow & Wootters, 2021). In particular, the popular federated learning (McMahan et al., 2017) belongs to the PS paradigm which has received increased attentions in recent years. We refer interested readers to (Kairouz et al., 2019) and references therein for a complete review. Most existing work is based on the centralized framework through synchronous communication scheme which often leads to communication congestion in the server. Alternatively, the PS can operate asynchronously, updating parameter vector immediately after it receives updates from a single worker. Although such an asynchronous update manner can increase system throughput, some workers may still operate on stale versions of the parameter vector and in some cases, even preventing convergence to the optimal model (Dai et al., 2018). More recently, several PS-based algorithms considering stale-synchronous model via adaptive worker participation have gained increased attention. See (Chen et al., 2016; Teng & Wood, 2018; Luo et al., 2019; Ruan et al., 2020; Xu et al., 2020; Yang et al., 2021) for details, among which (Xu et al., 2020) is perhaps the only work proposing to adaptively determine the number of fastest neighbor workers. However, most of them

focus on showing empirical results without convergence guarantees and without a rigorous analysis of how selection skew affects convergence speed. Moreover, none of them exploit the communication among workers, since they assume either a central server exists to coordinate workers (e.g., PS model) or all workers have an identical role (e.g., All-Reduce). Generalizing the PS schemes to our fully decentralized framework is a non-trivial and open problem because of the large scale distributed and heterogeneous nature of the training data.

## B  The Proposed Path Searching Procedure

For a given communication graph $\mathcal{G} = \{\mathcal{N}, \mathcal{E}\}$, `DSGD-AAU` establishes a strongly-connected graph such that there exists at least one path for arbitrary two nodes $i, j \in \mathcal{N}$ as quick as possible. We denote the set that contains all edges of such a strongly-connected graph as $\mathcal{P}$ and the set for all corresponding vertices as $\mathcal{V}$, which is essentially equal to $\mathcal{N}$. Then our key insight is that all edges in $\mathcal{P}$ have been visited at least once, i.e., all nodes in $\mathcal{V}$ share information with each other. To this end, we define a virtual epoch as the duration of the strongly-connect graph searching procedure and each epoch contains multiple iterations. Notice that the searching procedure can run multiple epochs until the convergence of parameters, and more importantly, every strongly-connected graph searching procedure is independent.

W.o.l.g., we consider one particular epoch of this procedure. The algorithm aims to establish at least one connected edge $(i, j)$ at each iteration so that $\mathcal{G}' := \{\mathcal{V}, \mathcal{P}\}$ is strongly connected and $\mathcal{V} = \mathcal{N}$ at the end of this epoch. At the beginning of the every epoch, $\mathcal{P}$ and $\mathcal{V}$ are reset to be empty, and all workers compute their local gradient simultaneously. Once one worker completes its local update, it sends its update to its neighbors and waits for collecting updates from its neighbors as well. If two workers $i, j$ ($i \in \mathcal{N}_j$ and $j \in \mathcal{N}_i$) successfully exchange their local updates, the edge $(i, j)$ is established only when $(i, j) \in \mathcal{E} \cap \{i \notin \mathcal{V} \text{ or } j \notin \mathcal{V}\}$. Then edge $(i, j)$ and nodes $i, j$ are added to $\mathcal{P}$ and $\mathcal{V}$, respectively, i.e., $\mathcal{P} \leftarrow \mathcal{P} \cup \{(i, j)\}$, $\mathcal{V} \leftarrow \mathcal{V} \cup \{i, j\}$, and all workers move to the next iteration. If the established edge satisfies $i \in \mathcal{V} \cap j \in \mathcal{V}$, then it will not be added to $\mathcal{P}$, and the current iteration continues until one such edge is established. The epoch ends when $\mathcal{G}'$ is strongly-connected and $\mathcal{V} = \mathcal{N}$.

We illustrate the `Pathsearch` algorithm as follows. In particular, at the each iteration $k$, denote the worker first finishes the local parameter update as $j_k$. Worker $j_k$ keeps idle until any one of the following two situations occurs. The first case is that one of worker $j_k$'s neighbor $i \in \mathcal{N}_{j_k}$ finishes the local parameter update and edge $(j_k, i) \in \mathcal{E} \cap (j_k, i) \notin \mathcal{P}$ and node $j_k \notin \mathcal{V} \bigcup i \notin \mathcal{V}$. Then store edge $(j_k, i)$ into $\mathcal{P}$ and nodes $\{i, j_k\}$ into $\mathcal{V}$. Then all workers move to the $(k+1)$-th iteration. Another case is that any other new edge $(i_1, j_1)$ between workers $i_1$ and $j_1$ other than $j_k$ is established such that edge $(i_1, j_1) \in \mathcal{E} \cap (i_1, j_1) \notin \mathcal{P}$ and node $j_1 \notin \mathcal{V} \bigcup i_1 \notin \mathcal{V}$. Then all workers move to the $(k+1)$-th iteration. The procedures are summarized in Algorithm 3.

---

**Algorithm 3** `Pathsearch` -logical view of arbitrary worker

**Input:** $\mathcal{G} = (\mathcal{N}, \mathcal{E})$, consensus sets $\mathcal{P}$ and $\mathcal{V}$, worker $j_k$.

1: Set $Flag \leftarrow 1$ and $\mathcal{N}_{j_k}(k) \leftarrow \phi$.
2: The worker $j_k$ from the set $\mathcal{N}$ finishes the local parameter update and let $\mathcal{N}_{j_k}(k) = \{j_k\}$;
3: **while** $Flag$ **do**
4:     Whenever a new worker $i \in \mathcal{N}_{j_k}$ finishes local parameter update, $\mathcal{N}_{j_k}(k) = \mathcal{N}_{j_k}(k) \cup \{i\}$;
5:     Check if any one of the following two situations occurs;
6:     **if** There exist two workers $i_1$ and $j_1$ in $\mathcal{N}_{j_k}(k)$ satisfying edge $(i_1, j_1) \in \mathcal{E} \cap (i_1, j_1) \notin \mathcal{P}$ and node $j_1 \notin \mathcal{V} \bigcup i_1 \notin \mathcal{V}$ **then**
7:         $\mathcal{P} \leftarrow \mathcal{P} \cup \{(i_1, j_1)\}$, $\mathcal{V} \leftarrow \mathcal{V} \cup \{i_1, j_1\}$;
8:         $Flag \leftarrow 0$;
9:     **else if** $\mathcal{P}$ and $\mathcal{V}$ at worker $j_k$ get consensus updated from neighbor workers **then**
10:        $Flag \leftarrow 0$;
11:    **end if**
12: **end while**

**Return:** $\mathcal{N}_{j_k}(k)$, consensus sets $\mathcal{P}$ and $\mathcal{V}$.

---

# C  Proofs of Main Result

In this section, we provide proofs of the theoretical results presented in the paper, i.e., the convergence for `DSGD-AAU` after a large number of iterations (Theorem 4.6).

## C.1  Proof of Theorem 4.6

*Proof.* We define $\mathbf{\Phi}_{k:s}$ as the product of consensus matrices from $\mathbf{P}(s)$ to $\mathbf{P}(k)$, i.e.,

$$\mathbf{\Phi}_{k:s} \triangleq \mathbf{P}(s)\mathbf{P}(s+1)\cdots\mathbf{P}(k),$$

and let $\Phi_{k,s}(i,j)$ be the $i$-th row $j$-th column element of $\mathbf{\Phi}_{k:s}$. Therefore, with fixed $\eta$, $\mathbf{w}_i(k)$ can be expressed in terms of $\mathbf{\Phi}$ as

$$\mathbf{w}_i(k+1) = \sum_{j=1}^{N} \Phi_{k,1}(i,j)\mathbf{w}_j(0) - \eta \sum_{r=1}^{k} \sum_{j=1}^{N} \Phi_{k,r}(i,j)g_i(\mathbf{w}_j(r-1)).$$

Next, we define an auxiliary variable $\bar{\mathbf{w}}(k) = \frac{\sum_j \mathbf{w}_j(k)}{N}$ satisfying

$$\bar{\mathbf{w}}(k) = \frac{1}{N}\sum_{j=1}^{N}\mathbf{w}_j(0) - \eta\sum_{r=1}^{k}\sum_{j=1}^{N}\frac{1}{N}g(\mathbf{w}_j(r-1)), \tag{8}$$

with the following relation holds

$$\bar{\mathbf{w}}(k+1) = \bar{\mathbf{w}}(k) - \frac{\eta}{N}\sum_{j=1}^{N}g_j(\mathbf{w}_j(k)). \tag{9}$$

Due to the gradient Lipschitz-continuous in Assumption 4.3, we have the following inequality with respect to $\bar{\mathbf{w}}(k)$ (Bottou et al., 2018)

$$
\begin{aligned}
\mathbb{E}\big[F(\bar{\mathbf{w}}(k+1))\big] \leq & \mathbb{E}\left[\nabla F(\bar{\mathbf{w}}(k))^{\intercal}(\bar{\mathbf{w}}(k+1)-\bar{\mathbf{w}}(k))\right] \\
& + \frac{L}{2}\mathbb{E}\left[\|\bar{\mathbf{w}}(k+1)-\bar{\mathbf{w}}(k)\|^2\right] + \mathbb{E}[F(\bar{\mathbf{w}}(k))] \\
= & \underbrace{\mathbb{E}\left[\nabla F(\bar{\mathbf{w}}(k))^{\intercal}((\bar{\mathbf{w}}(k+1) - \bar{\mathbf{w}}(k)) + \eta\nabla F(\bar{\mathbf{w}}(k)))\right]}_{C_1} \\
& + \underbrace{\frac{L}{2}\mathbb{E}\left[\|\bar{\mathbf{w}}(k+1) - \bar{\mathbf{w}}(k)\|^2\right] + \mathbb{E}[F(\bar{\mathbf{w}}(k))]}_{C_2} - \eta\|\nabla F(\bar{\mathbf{w}}(k))\|^2.
\end{aligned} \tag{10}
$$

As a result, we need to bound $C_1$ and $C_2$ in (10). The term $C_1$ can be bounded as follows

$$
\begin{aligned}
C_1 = & \mathbb{E}\left\langle \nabla F(\bar{\mathbf{w}}(k)), (\bar{\mathbf{w}}(k+1) - \bar{\mathbf{w}}(k)) + \eta\nabla F(\bar{\mathbf{w}}(k)) \right\rangle \\
\stackrel{(a_1)}{=} & \mathbb{E}\left\langle \nabla F(\bar{\mathbf{w}}(k)), \left[-\frac{\eta}{N}\sum_{j=1}^{N}g_j(\mathbf{w}_j(k)) + \frac{\eta}{N}\sum_{j=1}^{N}\nabla F_j(\bar{\mathbf{w}}(k))\right]\right\rangle \\
\stackrel{(a_2)}{=} & \mathbb{E}\left\langle \sqrt{\eta}\nabla F(\bar{\mathbf{w}}(k)), \left[-\frac{\sqrt{\eta}}{N}\sum_{j=1}^{N}(g_j(\mathbf{w}_j(k)) - \nabla F_j(\bar{\mathbf{w}}(k)))\right]\right\rangle \\
\stackrel{(a_3)}{=} & \frac{\eta}{2N^2}\mathbb{E}\left\|\sum_{j=1}^{N}(\nabla F_j(\mathbf{w}_j(k)) - \nabla F_j(\bar{\mathbf{w}}(k)))\right\|^2
\end{aligned}
$$

$$- \frac{\eta}{2N^2} \mathbb{E} \left\| \sum_{j=1}^{N} g_j(\mathbf{w}_j(k)) \right\|^2 + \frac{\eta}{2} \left\| \nabla F(\bar{\mathbf{w}}(k)) \right\|^2$$

$$\overset{(a_4)}{\leq} \underbrace{\frac{\eta}{2N} \sum_{j=1}^{N} \mathbb{E} \left\| \nabla F_j(\mathbf{w}_j(k)) - \nabla F_j(\bar{\mathbf{w}}(k)) \right\|^2}_{D_1}$$

$$- \frac{\eta}{2N^2} \mathbb{E} \left\| \sum_{j=1}^{N} g_j(\mathbf{w}_j(k)) \right\|^2 + \frac{\eta}{2} \left\| \nabla F(\bar{\mathbf{w}}(k)) \right\|^2, \tag{11}$$

where $(a_1)$ follows from (9) and the fact that $\frac{1}{N} \sum_{j=1}^{N} \nabla F_j(\bar{\mathbf{w}}(k)) = \nabla F(\bar{\mathbf{w}}(k))$; $(a_2)$ comes from a standard mathematical manipulation; the equality $(a_3)$ is based on the equation $\langle \mathbf{a}, \mathbf{b} \rangle = \frac{1}{2} \left[ \|\mathbf{a}\|^2 + \|\mathbf{b}\|^2 - \|\mathbf{a} - \mathbf{b}\|^2 \right]$ and $\mathbb{E}[g_j(\mathbf{w}_j(k))] = \nabla F_j(\mathbf{w}_j(k))$ from Assumption 4.4; and the last inequality $(a_4)$ is due to the well triangle-inequality $\left\| \sum_{i=1}^{N} \mathbf{a}_i \right\|^2 \leq N \sum_{i}^{N} \|\mathbf{a}_i\|^2$.

Hence, $D_1$ is bounded as follows

$$D_1 \leq \frac{\eta L^2}{2N} \sum_{j=1}^{N} \mathbb{E} \left[ \|\mathbf{w}_j(k) - \bar{\mathbf{w}}(k)\|^2 \right]$$

where the inequality directly comes from Assumption 4.3.

Substituting $D_1$ into $C_1$ yields

$$C_1 \leq \frac{\eta L^2}{2N} \sum_{j=1}^{N} \mathbb{E} \|\mathbf{w}_j(k) - \bar{\mathbf{w}}(k)\|^2 - \frac{\eta}{2N^2} \mathbb{E} \left\| \sum_{j=1}^{N} g_j(\mathbf{w}_j(k)) \right\|^2 + \frac{\eta}{2} \left\| \nabla F(\bar{\mathbf{w}}(k)) \right\|^2. \tag{12}$$

Next, we bound $C_2$ as

$$C_2 = \frac{L}{2} \mathbb{E} \left[ \|\bar{\mathbf{w}}(k+1) - \bar{\mathbf{w}}(k)\|^2 \right]$$

$$= \frac{\eta^2 L}{2N^2} \mathbb{E} \left[ \left\| \sum_{j=1}^{N} g_j(\mathbf{w}_j(k)) \right\|^2 \right]$$

$$\overset{(c_1)}{=} \frac{\eta^2 L}{2N^2} \left( \mathbb{E} \left[ \left\| \sum_{j=1}^{N} g_j(\mathbf{w}_j(k)) - \nabla F_j(\mathbf{w}_j(k)) \right\|^2 \right] + \left\| \mathbb{E} \sum_{j=1}^{N} g_j(\mathbf{w}_j(k)) \right\|^2 \right)$$

$$\overset{(c_2)}{\leq} \frac{\eta^2 L}{2N} \sigma_L^2 + \frac{\eta^2 L}{2N^2} \mathbb{E} \left\| \sum_{j=1}^{N} g_j(\mathbf{w}_j(k)) \right\|^2, \tag{13}$$

where $(c_1)$ is due to $\mathbb{E}[\|\mathbf{X}\|^2] = \mathrm{Var}[\mathbf{X}] + \|\mathbb{E}[\mathbf{X}]\|^2$ and $(c_2)$ follows the bounded variance in Assumption 4.5 and the fact that $\|\mathbb{E}[\mathbf{X}]\|^2 \leq \mathbb{E}[\|\mathbf{X}\|^2]$.

Substituting $C_1$ and $C_2$ into the inequality (10), we have

$$\mathbb{E}[F(\bar{\mathbf{w}}(k+1))]$$

$$\leq \mathbb{E}[F(\bar{\mathbf{w}}(k))] - \eta \|\nabla F(\bar{\mathbf{w}}(k))\|^2 + \frac{\eta}{2} \|\nabla F(\bar{\mathbf{w}}(k))\|^2 + \frac{\eta^2 L}{2N} \sigma_L^2$$

$$+ \frac{\eta^2 L}{2N^2} \mathbb{E} \left\| \sum_{j=1}^{N} g(\mathbf{w}_j(k)) \right\|^2 - \frac{\eta}{2N^2} \mathbb{E} \left\| \sum_{j=1}^{N} g(\mathbf{w}_j(k)) \right\|^2 + \frac{\eta L^2}{2N} \sum_{j=1}^{N} \mathbb{E} \left\| \mathbf{w}_j(k) - \bar{\mathbf{w}}(k) \right\|^2$$

$$\leq \mathbb{E}[F(\bar{\mathbf{w}}(k))] - \frac{\eta}{2} \|\nabla F(\bar{\mathbf{w}}(k))\|^2 + \frac{\eta^2 L - \eta}{2N^2} \mathbb{E} \left\| \sum_{j=1}^{N} g(\mathbf{w}_j(k)) \right\|^2$$

$$+ \frac{\eta L^2}{2N} \sum_{j=1}^{N} \mathbb{E} \left\| \mathbf{w}_j(k) - \bar{\mathbf{w}}(k) \right\|^2 + \frac{\eta^2 L}{2N} \sigma_L^2. \tag{14}$$

To characterize the convergence speed, the key boils down to bound $\mathbb{E}\|\mathbf{w}_j(k) - \bar{\mathbf{w}}(k)\|^2$. We bound it as follows.

$$\mathbb{E}\|\mathbf{w}_j(k) - \bar{\mathbf{w}}(k)\|^2$$

$$= \mathbb{E} \left\| \frac{1}{N} \sum_{i=1}^{N} \mathbf{w}_i(0) - \frac{\eta}{N} \sum_{s=1}^{k} \sum_{i=1}^{N} g_i(\mathbf{w}_i(s-1)) \right.$$

$$\left. - \sum_{i=1}^{N} \mathbf{w}_i(0) \Phi_{k:1}(j,i) + \eta \sum_{s=1}^{k} \sum_{i=1}^{N} g_i(\mathbf{w}_i(s-1)) \Phi_{k:s}(j,i) \right\|^2$$

$$= \mathbb{E} \left\| \sum_{i=1}^{N} \mathbf{w}_i(0)(\frac{1}{N} - \Phi_{k:1}(j,i)) - \eta \sum_{s=1}^{k} \sum_{i=1}^{N} g_i(\mathbf{w}_i(s-1))(\frac{1}{N} - \Phi_{k:s}(j,i)) \right\|^2$$

$$\overset{(b_1)}{\leq} \mathbb{E} \left\| \sum_{i=1}^{N} \mathbf{w}_i(0)(\frac{1}{N} - \Phi_{k:1}(i,j)) \right\|^2 + \mathbb{E} \left\| \eta \sum_{s=1}^{k} \sum_{i=1}^{N} g_i(\mathbf{w}_i(s-1))(\frac{1}{N} - \Phi_{k:s}(i,j)) \right\|^2$$

$$\overset{(b_2)}{\leq} \mathbb{E} \left\| \sum_{i=1}^{N} 2\mathbf{w}_i(0) \frac{1 + \beta^{-NB}}{1 - \beta^{NB}} (1 - \beta^{NB})^{(k-1)/NB} \right\|^2$$

$$+ \mathbb{E} \left\| \eta \sum_{s=1}^{k} \sum_{i=1}^{N} 2g(\mathbf{w}_i(s-1)) \frac{1 + \beta^{-NB}}{1 - \beta^{NB}} (1 - \beta^{NB})^{(k-s)/NB} \right\|^2$$

$$\overset{(b_3)}{=} 2\mathbb{E} \left\| \eta C \sum_{s=1}^{k} q^{(k-s)} \sum_{i=1}^{N} g_i(\mathbf{w}_i(s-1)) \right\|^2$$

$$= 2\mathbb{E} \left\| \eta C \sum_{s=1}^{k} q^{(k-s)} \sum_{i=1}^{N} (g_i(\mathbf{w}_i(s-1)) - \nabla F_i(\mathbf{w}_i(s-1))) + \eta C \sum_{s=1}^{k} q^{(k-s)} \sum_{i=1}^{N} \nabla F_i(\mathbf{w}_i(s-1)) \right\|^2$$

$$\overset{(b_4)}{\leq} 2\mathbb{E} \left\| \eta C \sum_{s=1}^{k} q^{(k-s)} \sum_{i=1}^{N} (g_i(\mathbf{w}_i(s-1)) - \nabla F_i(\mathbf{w}_i(s-1))) \right\|^2$$

$$+ \mathbb{E} 4\eta^2 C^2 \sum_{s=1}^{k} \sum_{s'=1}^{k} q^{(2k-s-s')} \left\| \sum_{i=1}^{N} (g_i(\mathbf{w}_i(s-1)) - \nabla F_i(\mathbf{w}_i(s-1))) \right\| \cdot \left\| \sum_{i=1}^{N} \nabla F_i(\mathbf{w}_i(s-1)) \right\|$$

$$+ 2\mathbb{E} \left\| \eta C \sum_{s=1}^{k} q^{(k-s)} \sum_{i=1}^{N} \nabla F_i(\mathbf{w}_i(s-1)) \right\|^2$$

$$\overset{(b_5)}{\leq} \mathbb{E} \frac{10\eta^2 C^2}{1-q} \sum_{s=1}^{k} q^{k-s} \left\| \sum_{i=1}^{N} (g_i(\mathbf{w}_i(s-1)) - \nabla F_i(\mathbf{w}_i(s-1))) \right\|^2$$

$$+ \mathbb{E}\frac{10\eta^2 C^2}{1-q}\sum_{s=1}^{k}q^{k-s}\left\|\sum_{i=1}^{N}\nabla F_i(\mathbf{w}_i(s-1))\right\|^2$$

$$\overset{(b_6)}{\leq}\frac{10\eta^2 C^2 N^2}{(1-q)^2}\sigma^2 + \frac{30\eta^2 C^2 N^2}{(1-q)^2}\varsigma^2 + \frac{30\eta^2 C^2 L^2}{1-q}\sum_{s=1}^{k}q^{k-s}\sum_{i=1}^{N}\mathbb{E}\|\mathbf{w}_i(s)-\mathbf{y}(s)\|^2$$

$$+ \frac{30\eta^2 C^2 N^2}{1-q}\sum_{s=1}^{k}q^{k-s}\mathbb{E}\|\nabla F(\mathbf{y}(s))\|^2, \tag{15}$$

where the inequality $(b_1)$ is due to the inequality $\|\mathbf{a}-\mathbf{b}\|^2 \leq \|\mathbf{a}\|^2 + \|\mathbf{b}\|^2$. $(b_2)$ holds according to Lemma C.3 (see the "Auxiliary Lemmas" Section below). W.l.o.g., we assume the initial term $\mathbf{w}_i(0), \forall i$ is small enough and we can neglect it. $(b_3)$ follows $C := 2 \cdot \frac{1+\beta^{-NB}}{1-\beta^{NB}}$ and $q := (1-\beta^{NB})^{1/NB}$. $(b_4)$ is due to $\|\mathbf{a}+\mathbf{b}\|^2 \leq \|\mathbf{a}\|^2 + \|\mathbf{b}\|^2 + 2\mathbf{a}\mathbf{b}$ and $(b_5)$ is some standard mathematical manipulation by leveraging the following inequality, i.e., for any $q \in (0,1)$ and non-negative sequence $\{\chi(s)\}_{s=0}^{k}$, it holds (Assran et al., 2019) $\sum_{k=0}^{K}\sum_{s=0}^{k}q^{k-s}\chi(s) \leq \frac{1}{1-\lambda}\sum_{s=0}^{K}\chi(s)$. $(b_6)$ holds due to Assumption 4.4, the inequality

$$\sum_{k=0}^{K}q^k\sum_{s=0}^{k}q^{k-s}\chi(s) \leq \sum_{k=0}^{K}\sum_{s=0}^{k}q^{2(k-s)}\chi(s) \leq \frac{1}{1-q^2}\sum_{s=0}^{K}\chi(s),$$

and the fact that

$$\mathbb{E}\|\nabla F_i(\mathbf{w}_i(k))\|^2 \leq \underbrace{3\mathbb{E}\|\nabla F_i(\mathbf{w}_i(k)) - \nabla F_i(\bar{\mathbf{w}}(k))\|^2}_{\text{Lipschitz continuous}} + \underbrace{3\mathbb{E}\|\nabla F_i(\bar{\mathbf{w}}(k)) - \nabla F(\bar{\mathbf{w}}(k))\|^2}_{\text{Bounded variance}}$$

$$+ 3\mathbb{E}\|\nabla F(\bar{\mathbf{w}}(k))\|^2$$

$$\leq 3L^2\mathbb{E}\|\mathbf{w}_i(k)-\bar{\mathbf{w}}(k)\|^2 + 3\varsigma^2 + 3\mathbb{E}\|\nabla F(\bar{\mathbf{w}}(k))\|^2.$$

Rearranging the order of each term in (14) and letting $L\eta \leq 1$, we have

$$\frac{\eta}{2}\|\nabla F(\bar{\mathbf{w}}(k))\|^2 \leq \mathbb{E}[F(\bar{\mathbf{w}}(k))] - \mathbb{E}[F(\bar{\mathbf{w}}(k+1))] + \frac{\eta L^2}{2N}\sum_{j=1}^{N}\mathbb{E}\|\mathbf{w}_j(k)-\bar{\mathbf{w}}(k)\|^2$$

$$+ \frac{L\eta^2}{2N}\sigma_L^2 + \frac{\eta}{2}\sigma_L^2 + \underbrace{\frac{\eta^2 L - \eta}{2N^2}\mathbb{E}\left\|\sum_{j=1}^{N}g(\mathbf{w}_j(k))\right\|^2}_{\leq 0}$$

$$\leq \mathbb{E}[F(\bar{\mathbf{w}}(k))] - \mathbb{E}[F(\bar{\mathbf{w}}(k+1))] + \frac{L\eta^2}{2N}\sigma_L^2$$

$$+ \frac{\eta L^2}{2N}\sum_{j=1}^{N}\mathbb{E}\|\mathbf{w}_j(k)-\bar{\mathbf{w}}(k)\|^2. \tag{16}$$

Since we have

$$\sum_{k=0}^{K-1}\sum_{j=1}^{N}\mathbb{E}\|\mathbf{w}_j(k)-\bar{\mathbf{w}}(k)\|^2$$

$$\leq \sum_{k=0}^{K-1}\left(\frac{30N\eta^2 C^2 L^2}{1-q}\sum_{s=0}^{k}q^{k-s}\sum_{i=1}^{N}\mathbb{E}\|\mathbf{w}_i(s)-\bar{\mathbf{w}}(s)\|^2\right.$$

$$\left.+ \frac{30\eta^2 C^2 N^3}{1-q}\sum_{s=1}^{k}q^{k-s}\mathbb{E}\|\nabla F(\bar{\mathbf{w}}(s))\|^2 + \frac{10\eta^2 C^2 N^3}{(1-q)^2}\sigma^2 + \frac{30\eta^2 C^2 N^2}{(1-q)^2}\varsigma^2\right)$$

$$\leq \left( \frac{30N\eta^2 C^2 L^2}{(1-q)^2} \sum_{k=0}^{K-1} \sum_{i=1}^{N} \mathbb{E}\|\mathbf{w}_i(k) - \bar{\mathbf{w}}(k)\|^2 \right.$$

$$\left. + \frac{30\eta^2 C^2 N^3}{(1-q)^2} \sum_{k=0}^{K-1} \mathbb{E}\|\nabla F(\bar{\mathbf{w}}(s))\|^2 + \frac{10K\eta^2 C^2 N^3}{(1-q)^2}\sigma^2 + \frac{30K\eta^2 C^2 N^2}{(1-q)^2}\varsigma^2 \right)$$

$$\leq \frac{30\eta^2 C^2 N^3}{(1-q)^2 - 30N\eta^2 C^2 L^2} \sum_{k=0}^{K-1} \mathbb{E}\|\nabla F(\bar{\mathbf{w}}(s))\|^2 + \frac{10K\eta^2 C^2 N^3}{(1-q)^2 - 30N\eta^2 C^2 L^2}\sigma^2$$

$$+ \frac{30K\eta^2 C^2 N^3}{(1-q)^2 - 30N\eta^2 C^2 L^2}\varsigma^2,$$

summing the recursion in (16) from the 0-th iteration to the $(K-1)$-th iteration yields

$$\sum_{k=0}^{K-1} \frac{\eta}{2}\|\nabla F(\bar{\mathbf{w}}(k))\|^2 \leq \mathbb{E}[F(\bar{\mathbf{w}}(0))] - \mathbb{E}[F(\bar{\mathbf{w}}(K))] + \frac{KL\eta^2}{2N}\sigma_L^2$$

$$+ \frac{\eta L^2}{2N} \sum_{k=0}^{K} \sum_{j=1}^{N} \mathbb{E}\|\mathbf{w}_j(k) - \bar{\mathbf{w}}(k)\|^2$$

$$\leq \mathbb{E}[F(\bar{\mathbf{w}}(0))] - \mathbb{E}[F(\bar{\mathbf{w}}(K))] + \frac{KL\eta^2}{2N}\sigma_L^2$$

$$+ \frac{15\eta^3 C^2 N^2 L^2}{(1-q)^2 - 30N\eta^2 C^2 L^2} \sum_{k=0}^{K-1} \mathbb{E}\|\nabla F(\bar{\mathbf{w}}(k))\|^2$$

$$+ \frac{5K\eta^3 C^2 N^2 L^2}{(1-q)^2 - 30N\eta^2 C^2 L^2}\sigma^2 + \frac{15K\eta^3 C^2 N^2 L^2}{(1-q)^2 - 30N\eta^2 C^2 L^2}\varsigma^2. \tag{17}$$

Hence we have

$$\left( \frac{\eta}{2} - \frac{15\eta^3 C^2 N^2 L^2}{(1-q)^2 - 30N\eta^2 C^2 L^2} \right) \sum_{k=0}^{K-1} \|\nabla F(\bar{\mathbf{w}}(k))\|^2$$

$$\leq \mathbb{E}[F(\bar{\mathbf{w}}(0))] - \mathbb{E}[F(\bar{\mathbf{w}}(K))] + \frac{KL\eta^2}{2N}\sigma_L^2$$

$$+ \frac{5K\eta^3 C^2 N^2 L^2}{(1-q)^2 - 30N\eta^2 C^2 L^2}\sigma_L^2 + \frac{15K\eta^3 C^2 N^2 L^2}{(1-q)^2 - 30N\eta^2 C^2 L^2}\varsigma^2. \tag{18}$$

Let $\eta \leq \min\left( \sqrt{\frac{(1-q)^2}{30C^2 L^2 N} + \frac{9N^4}{16}} - \frac{3N^2}{4}, 1/L \right)$, then we have

$$\frac{\eta}{2} - \frac{15\eta^3 C^2 N^2 L^2}{(1-q)^2 - 30N\eta^2 C^2 L^2} \geq \eta/6,$$

and

$$(1-q)^2 - 30N\eta^2 C^2 L^2 \geq 45\eta C^2 N^3 L^2.$$

Hence we have

$$\frac{\eta}{6} \sum_{k=0}^{K-1} \|\nabla F(\bar{\mathbf{w}}(k))\|^2 \leq \mathbb{E}[F(\bar{\mathbf{w}}(0))] - \mathbb{E}[F(\bar{\mathbf{w}}(K))] + \frac{KL\eta^2}{2N}\sigma_L^2 + \frac{K\eta^2}{9N}\sigma_L^2 + \frac{K\eta^2}{3N}\varsigma^2. \tag{19}$$

Diving both sides with $\eta K/6$, we have

$$\frac{1}{K} \sum_{k=0}^{K-1} \|\nabla F(\bar{\mathbf{w}}(k))\|^2 \leq \mathbb{E}\frac{6[F(\bar{\mathbf{w}}(0))] - \mathbb{E}[F(\bar{\mathbf{w}}(K))]}{\eta K} + \frac{9L\eta + 2\eta}{3N}\sigma_L^2 + \frac{2\eta}{N}\varsigma^2. \tag{20}$$

This completes the proof.

$\square$

## C.2 Proof of Corollary 4.8

**Corollary C.1.** *Let $\eta = \sqrt{N/K}$. The convergence rate of Algorithm 1 is $\mathcal{O}\left(\frac{1}{\sqrt{NK}}\right)$.*

*Proof.* The desired results is yield by substituting $\eta = \sqrt{N/K}$ back in to (7) in Theorem 4.6. □

## C.3 Auxiliary Lemmas

We provide the following auxiliary lemmas which are used in our proofs. We omit the proofs of these lemmas for the ease of exposition and refer interested readers to (Xiao et al., 2006) and (Nedic & Ozdaglar, 2009) for details.

**Lemma C.2** (Theorem 2 in (Xiao et al., 2006)). *Assume that $\mathbf{P}(k)$ is doubly stochastic for all $k$. We denoted the limit matrix of $\Phi_{k,s} = \mathbf{P}(s)\mathbf{P}(s+1)\ldots\mathbf{P}(k)$ as $\Phi_s \triangleq \lim_{k\to\infty} \Phi_{k,s}$ for notational simplicity. Then, the entries $\Phi_{k,s}(i,j)$ converges to $1/N$ as $k$ goes to $\infty$ with a geometric rate. The limit matrix $\Phi_s$ is doubly stochastic and correspond to a uniform steady distribution for all $s$, i.e.,*

$$\Phi_s = \frac{1}{N}\mathbf{1}\mathbf{1}^T.$$

**Lemma C.3** (Lemma 4 in (Nedic & Ozdaglar, 2009)). *Assume that $\mathbf{P}(k)$ is doubly stochastic for all $k$. Under Assumption 2, the difference between $1/N$ and any element of $\Phi_{k,s} = \mathbf{P}(s)\mathbf{P}(s+1)\ldots\mathbf{P}(k)$ can be bounded by*

$$\left|\frac{1}{N} - \Phi_{k,s}(i,j)\right| \leq 2\frac{(1+\beta^{-NB})}{1-\beta^{NB}}(1-\beta^{NB})^{(k-s)/NB}, \tag{21}$$

*where $\beta$ is the smallest positive value of all consensus matrices, i.e., $\beta = \arg\min P_{i,j}(k), \forall k$ with $P_{i,j}(k) > 0, \forall i, j$.*

## C.4 Assumption on Adaptive Asynchronous Updates

We note that we focus on stragglers caused by heterogeneous computation capabilities among workers as in (Ananthanarayanan et al., 2013; Ho et al., 2013; Lian et al., 2017; 2018; Luo et al., 2020), while the communication bandwidth is not a bottleneck. Hence the major time consumption comes from the calculation of gradients. First, we prove that `DSGD-AAU` achieves a linear speedup for convergence in terms of iterations. In such case, the communication does not affect the result with respect to the convergence iteration. However, the communication time will impact the wall-clock time for convergence. To the best of knowledge, Prague (Luo et al., 2020) is perhaps the first work to separately measure the computation time and communication time by running different number of workers and found that the communication time is much smaller and dominated by the computation time. We further explore this and estimate the communication time in our system given the network speed. For instance, in our system, the network speed is about 20GB/s given the hardware of our server (memory card), and the communication time only accounts for 0.14%-4% of the total time measured in communication and computation time under the CIFAR-10 dataset. Moreover, our convergence time in the numerical evaluation considers the total time of the whole process, which includes both computation and communication time. Exactly same thing has been done in Prague (Luo et al., 2020), AD-PSGD (Lian et al., 2018), and Li et al. (Li et al., 2020a;b).

# D  Additional Experimental Results

In this section, we provide the experiment details, parameter settings and some additional experimental results of the experiment setting in Section 5.

**Dataset and Model.** We evaluate the performance of different algorithms on both the classification tasks and next-character prediction tasks. For the classification tasks, we use CIFAR-10 (Krizhevsky et al., 2009),

MNIST (LeCun et al., 1998) and Tiny-ImageNet (Le & Yang, 2015; Russakovsky et al., 2015) as the evaluation datasets for both i.i.d. and non-i.i.d. versions. However, we mainly focus on the non-i.i.d. versions since data are generated locally at the workers based on their circumstances in real-world systems (see Section 3). For the task of next-character prediction, we use the dataset of *The Complete Works of William Shakespeare* (McMahan et al., 2017) (Shakespeare).

The CIFAR-10 dataset consists of $60,000$ $32 \times 32$ color images in 10 classes where $50,000$ samples are for training and the other $10,000$ samples for testing. The MNIST dataset contains handwritten digits with $60,000$ samples for training and $10,000$ samples for testing. The Tiny-ImageNet (Le & Yang, 2015) has a total of 200 classes with 500 images each class for training and 50 for testing, which is drawn from the ImageNet dataset (Russakovsky et al., 2015) and down-sampled to $64 \times 64$. For MNIST, we split the data based on the digits they contain in the dataset, i.e., we distribute the data to all workers such that each worker contains only a certain class of digits with the same number of training/testing samples (e.g., same ideas were used in Yang et al. (Yang et al., 2021)). We also use the idea suggested by McMahan et al. (McMahan et al., 2017) and Yang et al. (Yang et al., 2021) to deal with CIFAR-10 dataset. We use $N = 128$ workers as an example. We sort the data by label of different classes (10 classes in total), and split each class into $N/2 = 64$ pieces. Each worker will randomly select 5 classes of data partitions to forms its local dataset.

For the classification tasks, we consider four representative models: a fully-connected neural network with 2 hidden layers (2NN), ResNet-18 (He et al., 2016), VGG-13 (Simonyan & Zisserman, 2015) and AlexNet (Krizhevsky et al., 2012), as Table 3, Table 4, Table 5 and Table 7, respectively. For the next-character prediction task, we use an LSTM language model as Table 6 as in (Kim et al., 2016).

| Parameter | Shape | Layer hyper-parameter |
|---|---|---|
| layer1.fc1.weight | $3072 \times 256$ | N/A |
| layer1.fc1.bias | 256 | N/A |
| layer2.fc2.weight | $256 \times 256$ | N/A |
| layer2.fc2.bias | 256 | N/A |
| layer3.fc3.weight | $256 \times 10$ | N/A |
| layer3.fc3.bias | 10 | N/A |

Table 3: Detailed information of the 2NN architecture used in our experiments. All non-linear activation function in this architecture is ReLU. The shapes for convolution layers follow $(C_{in}, C_{out}, c, c)$.

| Parameter | Shape | Layer hyper-parameter |
|---|---|---|
| layer1.conv1.weight | $3 \times 3, 64$ | stride:1; padding: 1 |
| layer1.conv1.bias | 64 | N/A |
| batchnorm2d | 64 | N/A |
| layer2.conv2 | $\begin{bmatrix} 3 \times 3, & 64 \\ 3 \times 3, & 64 \end{bmatrix} \times 2$ | stride:1; padding: 1 |
| layer3.conv3 | $\begin{bmatrix} 3 \times 3, & 128 \\ 3 \times 3, & 128 \end{bmatrix} \times 2$ | stride:1; padding: 1 |
| layer4.conv4 | $\begin{bmatrix} 3 \times 3, & 256 \\ 3 \times 3, & 256 \end{bmatrix} \times 2$ | stride:1; padding: 1 |
| layer5.conv5 | $\begin{bmatrix} 3 \times 3, & 512 \\ 3 \times 3, & 512 \end{bmatrix} \times 2$ | stride:1; padding: 1 |
| pooling.avg | N/A | N/A |
| layer6.fc6.weight | $512 \times 10$ | N/A |
| layer6.fc6.bias | 10 | N/A |

Table 4: Detailed information of the ResNet-18 architecture used in our experiments. All non-linear activation function in this architecture is ReLU. The shapes for convolution layers follow $(C_{in}, C_{out}, c, c)$.

| Parameter | Shape | Layer hyper-parameter |
|---|---|---|
| layer1.conv1.weight | $3 \times 64 \times 3 \times 3$ | stride:1; padding: 1 |
| layer1.conv1.bias | 64 | N/A |
| pooling.max | N/A | kernel size:2; stride: 2 |
| layer2.conv2.weight | $32 \times 64 \times 3 \times 3$ | stride:1; padding: 1 |
| layer2.conv2.bias | 64 | N/A |
| pooling.max | N/A | kernel size:2; stride: 2 |
| layer3.conv3.weight | $64 \times 128 \times 3 \times 3$ | stride:1; padding: 1 |
| layer3.conv3.bias | 128 | N/A |
| layer4.conv4.weight | $128 \times 128 \times 3 \times 3$ | stride:1; padding: 1 |
| layer4.conv4.bias | 128 | N/A |
| pooling.max | N/A | kernel size:2; stride: 2 |
| layer5.conv5.weight | $128 \times 256 \times 3 \times 3$ | stride:1; padding: 1 |
| layer5.conv5.bias | 256 | N/A |
| layer6.conv6.weight | $256 \times 256 \times 3 \times 3$ | stride:1; padding: 1 |
| layer6.conv6.bias | 256 | N/A |
| pooling.max | N/A | kernel size:2; stride: 2 |
| layer7.conv7.weight | $256 \times 512 \times 3 \times 3$ | stride:1; padding: 1 |
| layer7.conv7.bias | 512 | N/A |
| layer8.conv8.weight | $512 \times 512 \times 3 \times 3$ | stride:1; padding: 1 |
| layer8.conv8.bias | 512 | N/A |
| pooling.max | N/A | kernel size:2; stride: 2 |
| layer9.conv9.weight | $512 \times 512 \times 3 \times 3$ | stride:1; padding: 1 |
| layer9.conv9.bias | 512 | N/A |
| layer10.conv10.weight | $512 \times 512 \times 3 \times 3$ | stride:1; padding: 1 |
| layer10.conv10.bias | 512 | N/A |
| pooling.max | N/A | kernel size:2; stride: 2 |
| dropout | N/A | p=20% |
| layer11.fc11.weight | $4096 \times 512$ | N/A |
| layer11.fc11.bias | 512 | N/A |
| layer12.fc12.weight | $512 \times 512$ | N/A |
| layer12.fc12.bias | 512 | N/A |
| dropout | N/A | p=20% |
| layer13.fc13.weight | $512 \times 10$ | N/A |
| layer13.fc13.bias | 10 | N/A |

Table 5: Detailed information of the VGG-13 architecture used in our experiments. All non-linear activation function in this architecture is ReLU. The shapes for convolution layers follow $(C_{in}, C_{out}, c, c)$.

| Parameter | Shape | Layer hyper-parameter |
|---|---|---|
| layer1.embeding | $80 \times 256$ | N/A |
| layer2.lstm | $256 \times 512$ | num_layers=2, batch_first=True |
| dropout | N/A | p=5% |
| layer3.fc.weight | $512 \times 80$ | N/A |
| layer3.fc.bias | 80 | N/A |

Table 6: Detailed information of the LSTM model used in our experiments.

| Parameter | Shape | Layer hyper-parameter |
|---|---|---|
| layer1.conv1.weight | $3 \times 64 \times 3 \times 3$ | stride:2; padding: 1 |
| layer1.conv1.bias | 64 | N/A |
| pooling.max | N/A | kernel size:2; stride: 2 |
| layer2.conv2.weight | $64 \times 192 \times 3 \times 3$ | stride:1; padding: 1 |
| layer2.conv2.bias | 64 | N/A |
| pooling.max | N/A | kernel size:2; stride: 2 |
| layer3.conv3.weight | $192 \times 384 \times 3 \times 3$ | stride:1; padding: 1 |
| layer3.conv3.bias | 128 | N/A |
| layer4.conv4.weight | $384 \times 256 \times 3 \times 3$ | stride:1; padding: 1 |
| layer4.conv4.bias | 128 | N/A |
| layer5.conv5.weight | $256 \times 256 \times 3 \times 3$ | stride:1; padding: 1 |
| layer5.conv5.bias | 256 | N/A |
| pooling.max | N/A | kernel size:2; stride: 2 |
| dropout | N/A | p=20% |
| layer6.fc6.weight | $1024 \times 4096$ | N/A |
| layer6.fc6.bias | 512 | N/A |
| dropout | N/A | p=20% |
| layer7.fc7.weight | $4096 \times 4096$ | N/A |
| layer7.fc7.bias | 512 | N/A |
| layer8.fc8.weight | $4096 \times 10$ | N/A |
| layer8.fc8.bias | 10 | N/A |

Table 7: Detailed information of the AlexNet architecture used in our experiments. All non-linear activation function in this architecture is ReLU. The shapes for convolution layers follow $(C_{in}, C_{out}, c, c)$.

We consider a network with 32, 64, 128 and 256 workers and randomly generate a connected graph for evaluation. The loss functions we consider are the cross-entropy one. Our experiments are running using the following setups. Model: AMD TRX40 Motherboard. System: Ubuntu 20.04. CPU: AMD Ryzen threadripper 3960x 24-core processor x 48. GPU: NVIDIA RTX A6000. Network Speed: 20GB/s. Language: Python3, Tensorflow 2.0.

| Dataset | Model | AGP | AD-PSGD | Prague | DSGD-AAU |
|---|---|---|---|---|---|
| | 2-NN | 43.87% | 43.5% | 44.5% | **45.4%** |
| CIFAR-10 | AlexNet | 52.85% | 49.5% | 56.3% | **58.6%** |
| | VGG-13 | 59.41% | 56.3% | 63.5% | **67.1%** |
| | ResNet-18 | 76.25% | 73.7% | 77.3% | **79.8%** |
| | 2-NN | 95.75% | 95.56% | 95.86% | **95.98%** |
| MNIST | AlexNet | 95.15% | 95.12% | 95.47% | **95.58%** |
| | VGG-13 | 95.97% | 95.38% | 96.15% | **96.27%** |
| | ResNet-18 | 97.31% | 97.19% | 97.33% | **97.43%** |
| Tiny-ImageNet | ResNet-18 | 41.81% | 42.39% | 45.28% | **46.21%** |
| Shakespeare | LSTM | 54.88% | 54.65% | 55.72% | **56.22%** |

Table 8: The test accuracy of different models using different non-i.i.d. datasets with 128 workers.

| Dataset (Model) | # | AGP | AD-PSGD | Prague | DSGD-AAU |
|---|---|---|---|---|---|
| CIFAR-10 (ResNet-18) | 32 | 67.19% | 55.67% | 60.22% | **71.56%** |
| | 64 | 73.01% | 63.09% | 69.55% | **77.34%** |
| | 128 | 75.18% | 68.64% | 74.72% | **78.58%** |
| | 256 | 75.32% | 73.5% | 77.14% | **78.76%** |
| MNIST (ResNet-18) | 32 | 96.03% | 94.84% | 95.43% | **96.61%** |
| | 64 | 96.82% | 95.43% | 96.35% | **97.01%** |
| | 128 | 96.99% | 96.14% | 96.43% | **97.12%** |
| | 256 | 97.04% | 96.15% | 96.45% | **97.15%** |
| Tiny-ImageNet (ResNet-18) | 32 | 38.62% | 36.8% | 39.54% | **41.71%** |
| | 64 | 39.64% | 39.04% | 41.9% | **44.13%** |
| | 128 | 41.30% | 41.32% | 43.72% | **44.88%** |
| | 256 | 42.05% | 43.02% | 43.79% | **45.18%** |
| Shakespeare (LSTM) | 32 | 48.72% | 46.95% | 48.09% | **50.25%** |
| | 64 | 51.53% | 49.16% | 50.80% | **53.25%** |
| | 128 | 52.88% | 50.79% | 52.42% | **54.26%** |
| | 256 | 53.03% | 52.23% | 53.14% | **54.29%** |

Table 9: The test accuracy of ResNet-18 on non-i.i.d. CIFAR-10, MNIST, and Tiny-ImageNet when trained for 50 seconds, 10 seconds and 30 seconds, respectively; and the test accuracy of LSTM on non-i.i.d. Shakespeare when trained for 30 seconds.

**Test Accuracy.** Complementary to Table 1 in the main paper, we summarize the test accuracy of different models using different non-i.i.d. datasets with 128 workers in Table 8. Again, we observe that DSGD-AAU consistently outperforms all state-of-the-art baselines in consideration.

Complementary to Table 2 in the main paper, we summarize the test accuracy of different models over different non-i.i.d. datasets when trained for a limited time in Table 9 with different number of workers. For example, for the image classification tasks, we train ResNet-18 for 50 seconds on non-i.i.d. CIFAR-10, 10 seconds on non-i.i.d. MNIST, and 50 seconds for non-i.i.d. Tiny-ImageNet. For the next-character prediction task, we train LSTM on Shakespeare for 30 seconds. Again, we observe that for a limited training time, DSGD-AAU consistently achieves a higher test accuracy compared to AGP, AD-PSGD and Prague.

Complementary to Tables 8 and 9, the corresponding results when trained on the i.i.d. version of datasets are presented in Tables 10 and 11, respectively. Similar observations can be made and hence we omit the discussions here.

| Dataset (IID) | Model | AGP | AD-PSGD | Prague | DSGD-AAU |
|---|---|---|---|---|---|
| CIFAR-10 | 2-NN | 45.35% | 45.66% | 46.48% | **46.95%** |
| | AlexNet | 53.85% | 55.9% | 59.84% | **61.01%** |
| | VGG-13 | 59.51% | 63.12% | 66.62% | **68.26%** |
| | ResNet-18 | 78.61% | 78.24% | 81.47% | **82.91%** |
| MNIST | 2-NN | 96.55% | 96.96% | 97.25% | **97.27%** |
| | AlexNet | 95.87% | 96.56% | 96.89% | **96.93%** |
| | VGG-13 | 96.38% | 96.97% | 97.44% | **97.62%** |
| | ResNet-18 | 97.84% | 98.22% | 98.48% | **98.52%** |
| Tiny-ImageNet | ResNet-18 | 46.08% | 46.27% | 48.09% | **48.5%** |
| Shakespeare | LSTM | 54.88% | 54.65% | 55.72% | **56.22%** |

Table 10: The test accuracy of different models using different i.i.d. datasets with 128 workers.

| Dataset (Model) IID | # | AGP | AD-PSGD | Prague | DSGD-AAU |
|---|---|---|---|---|---|
| CIFAR-10 (ResNet-18) | 32 | 68.95% | 62.59% | 65.19% | **75.53%** |
| | 64 | 74.26% | 68.40% | 74.91% | **80.46%** |
| | 128 | 77.08% | 74.01% | 79.19% | **81.43%** |
| | 256 | 77.17% | 77.47% | 80.06% | **81.96%** |
| MNIST (ResNet-18)-10 | 32 | 96.93% | 96.40% | 96.89% | **97.36%** |
| | 64 | 97.38% | 96.85% | 97.52% | **98.06%** |
| | 128 | 97.47% | 97.13% | 97.57% | **98.16%** |
| | 256 | 97.56% | 97.23% | 97.67% | **98.22%** |
| Tiny-ImageNet (ResNet-18)-50 | 32 | 42.81% | 40.17% | 43.09% | **45.13%** |
| | 64 | 44.08% | 42.66% | 45.24% | **47.30%** |
| | 128 | 44.51% | 44.15% | 46.22% | **47.63%** |
| | 256 | 44.52% | 45.46% | 46.31% | **47.77%** |
| Shakespeare (LSTM)-30 | 32 | 48.72% | 46.95% | 48.09% | **50.25%** |
| | 64 | 51.53% | 49.16% | 50.80% | **53.25%** |
| | 128 | 52.88% | 50.79% | 52.42% | **54.26%** |
| | 256 | 53.03% | 52.23% | 53.14% | **54.29%** |

Table 11: The test accuracy of ResNet-18 on i.i.d. CIFAR-10, MNIST, and Tiny-ImageNet when trained for 50 seconds, 10 seconds and 30 seconds, respectively; and the test accuracy of LSTM on Shakespeare when trained for 30 seconds.

**Speedup and Communication.** Complementary to Figure 5 in the main paper, we present the corresponding results for 2-NN, AlexNet, and VGG-13 on non-i.i.d. partitioned CIFAR-10 in Figures 6, 7 and 8, respectively. Similarly, we presented the results on other datasets (i.e., MNIST, Tiny-ImageNet and Shakespeare) in Figures 9, 10, 11, 12, 13 and 14, respectively, using different models. Again, we observe that `DSGD-AAU` consistently outperforms baseline methods and achieves the best speedup at no cost of additional communication across different models and different datasets.

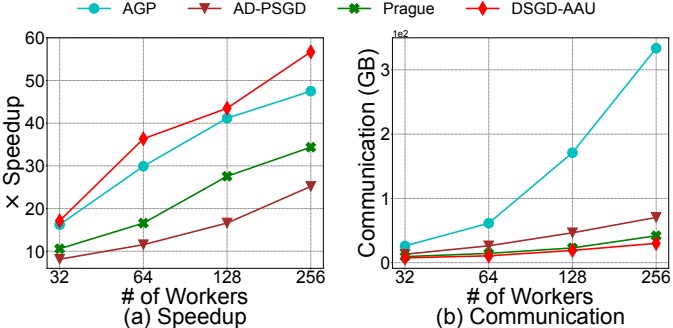

Figure 6: Speedup and communication for 2-NN on non-i.i.d. CIFAR-10 with different number of workers.

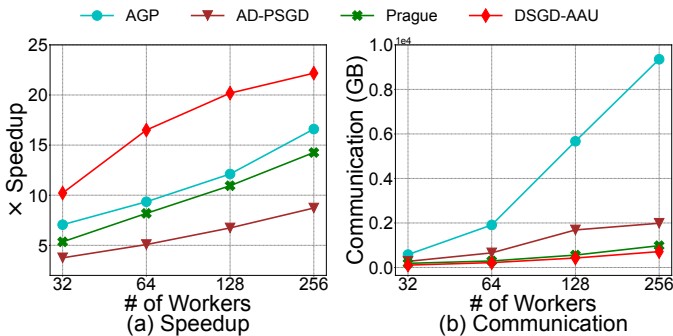

Figure 7: Speedup and communication for AlexNet on non-i.i.d. CIFAR-10 with different number of workers.

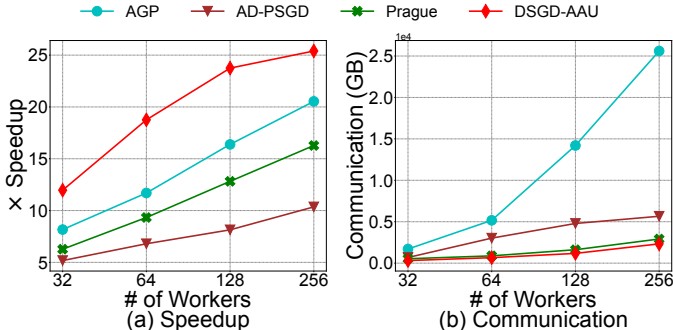

Figure 8: Speedup and communication for VGG-13 on non-i.i.d. CIFAR-10 with different number of workers.

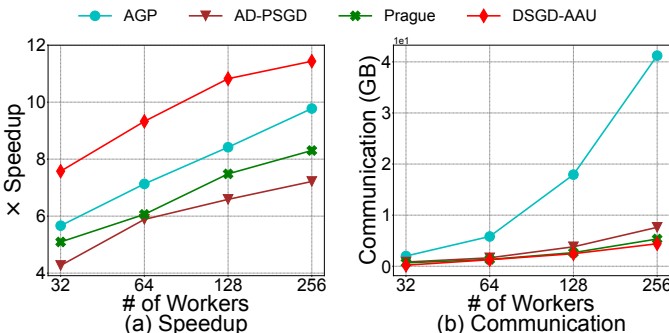

Figure 9: Speedup and communication for 2-NN on non-i.i.d. MNIST with different number of workers.

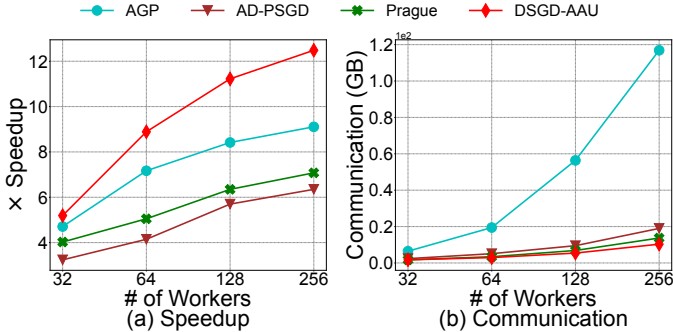

Figure 10: Speedup and communication for AlexNet on non-i.i.d. MNIST with different number of workers.

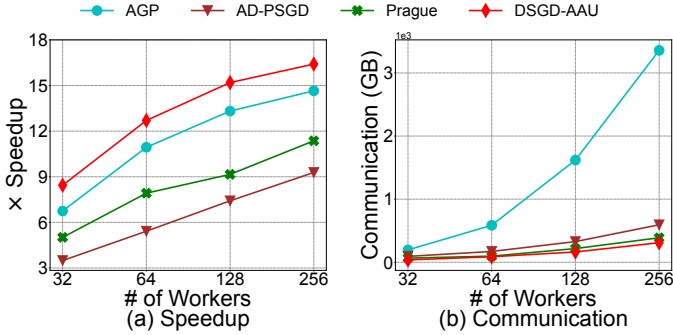

Figure 11: Speedup and communication for VGG-13 on non-i.i.d. MNIST with different number of workers.

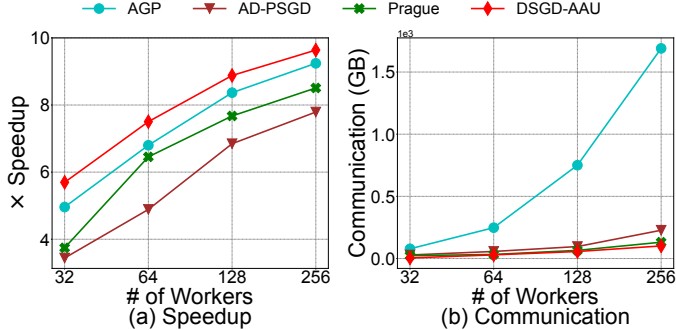

Figure 12: Speedup and communication for ResNet-18 on non-i.i.d. MNIST with different number of workers.

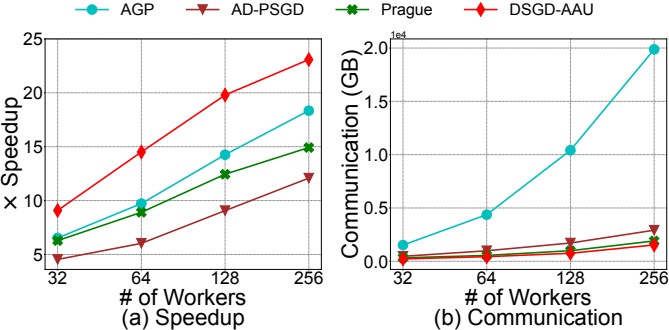

Figure 13: Speedup and communication for ResNet-18 on non-i.i.d. Tiny-ImageNet with different number of workers.

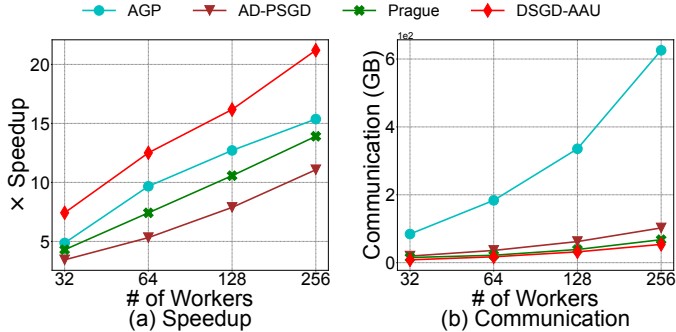

Figure 14: Speedup and communication for LSTM on Shakespeare with different number of workers.

**Ablation Study.** Complementary to discussions in Section 5, we consider two key hyperparameters in decentralized ML optimization problems, i.e., the batch size and the portion of stragglers. We now evaluate the their impact on the performance of VGG-13 using non-i.i.d. CIFAR-10 dataset with 128 workers. The final test accuracy is reported in Figure 15(a), while the test accuracy when trained VGG-13 for 50 seconds is reported in Figure 16(a). Based on these results, we find that the batch size of 128 is a proper value and use it throughout our experiments.

Furthermore, we investigate the impact of the portion of stragglers. Specifically, we consider the case that a work can become a straggler in one iteration/round with probability "P" and we call this the "straggler probability". We vary the straggler rate from 5% to 40%. The final test accuracy is reported in Figure 15(b), while the test accuracy when trained VGG-13 for 50 seconds is reported in Figure 16(b). On the one hand, we observe that as the straggler rate increases, the performance of all algorithms degrade, which is quite intuitive. On the other hand, we observe that our `DSGD-AAU` consistently outperforms state of the arts over all cases. We choose the straggler rate to be 10% throughout our experiments.

Finally, we investigate the impact of "slow down" for stragglers since we randomly select workers as stragglers in each iteration, e.g., the computing speed of the straggler is slowed down in the iteration. We vary the slow down from 5× to 40×. The final test accuracy is reported in Figure 15(c), while the test accuracy when trained VGG-13 for 50 seconds is reported in Figure 16(c). Again, we observe that our `DSGD-AAU` consistently outperforms state of the arts over all cases and we choose the slow down to be 10× in our experiments.

Similar observations can be made when considering the i.i.d. CIFAR-10, which is shown in Figures 17 and 18, as well as other models and datasets. In particular, the heterogeneity of non-i.i.d. datasets among different workers is quantified via a universal bound $\varsigma$ (Assumption 4.5). This general framework captures the i.i.d. dataset case, i.e., the i.i.d. datasets refer to a special case where $\varsigma = 0$. This is reflected in the convergence bound (Theorem 4.6) with a non-vanishing constant term $\frac{2\eta}{N}\varsigma^2$. Especially, for the i.i.d. case, we present the test accuracy of different models using different i.i.d. datasets with 128 workers in Table 10. We also summarize the test accuracy of different models over different i.i.d. datasets when trained for a limited time in Table 11 with different numbers of workers. These results show that DSGD-AAU consistently achieves a higher test accuracy compared to other methods, both for i.i.d. and non-i.i.d. datasets.

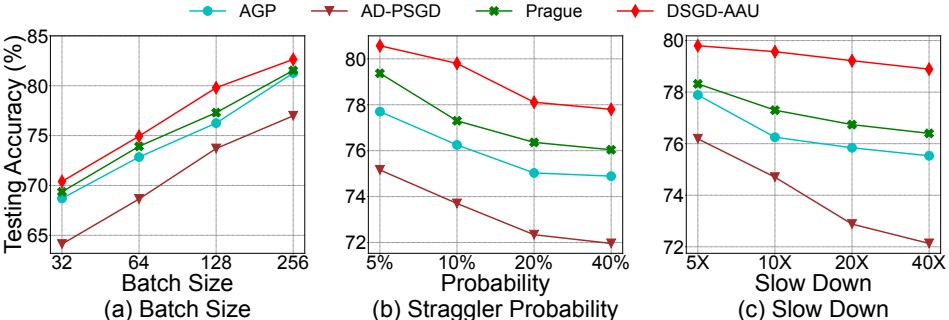

Figure 15: Ablation study for VGG-13 on non-i.i.d. CIFAR-10.

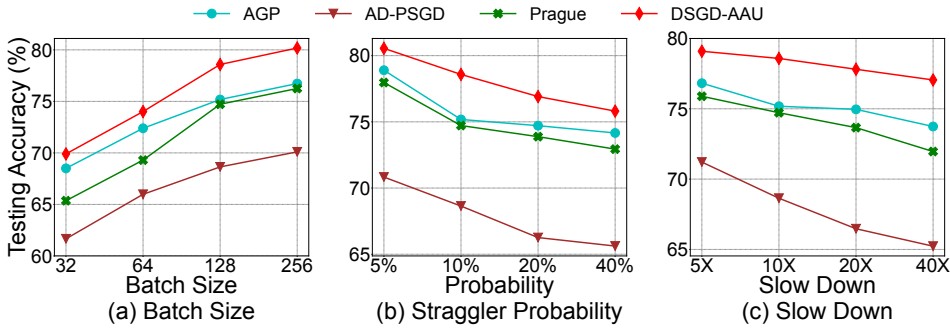

Figure 16: Ablation study for VGG-13 when trained for 50 seconds on non-i.i.d. CIFAR-10.

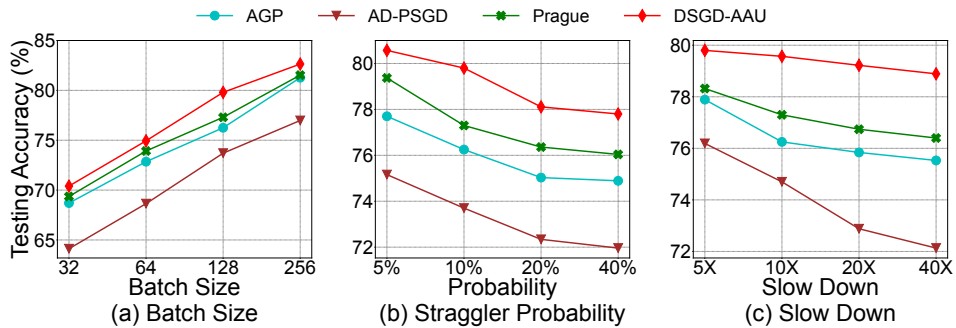

Figure 17: Ablation study for VGG-13 on i.i.d. CIFAR-10.

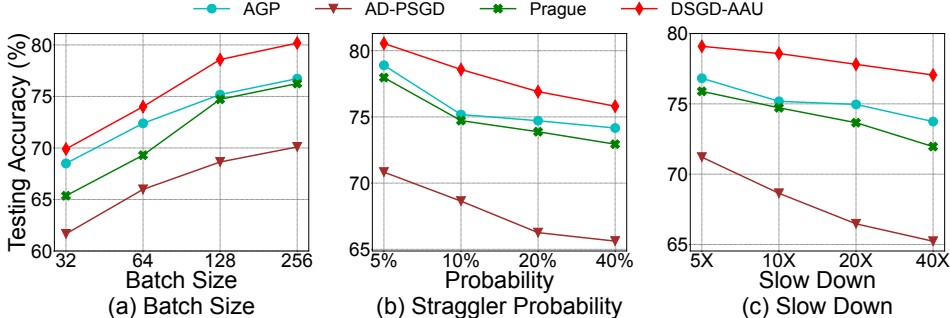

Figure 18: Ablation study for VGG-13 when trained for 50 seconds on i.i.d. CIFAR-10.

