# OpenReview forum: "Straggler-Resilient Decentralized Learning via Adaptive Asynchronous Updates"
_TMLR — Rejected by TMLR_

### Review · Reviewer_Tbdh · 2023-07-09

**Summary Of Contributions:**

This paper proposes a fully decentralized algorithm for large-scale training of machine learning models, called DSGD-AAU. The algorithm allows each worker to communicate with a subset of its neighbors, instead of waiting for all of them, to mitigate the effect of stragglers. Since the algorithm does not need to perform local update immediately using stale information, it also mitigates the effect of gradient staleness. The authors propose a Path Searching Procedure to establish a strongly-connected graph in each iteration. The paper shows that DSGD-AAU achieves a linear speedup with the number of workers and provides theoretical guarantee for its convergence rate with respect to communication rounds. The paper also presents experimental results on several datasets and deep neural network models to demonstrate the effectiveness and efficiency of DSGD-AAU.

**Audience:**

Yes

**Claims And Evidence:**

Yes

**Requested Changes:**

Please add more description on how Algorithm 2 implements the idea of selecting a subset of the fastest computational workers $\mathcal{N}(k)$. Also it would be nice if the authors could provide more theoretical justifications for mitigating the stragglers issue without suffering from the gradient staleness.

**Strengths And Weaknesses:**

Strengths:
- This paper is clearly written, with clear motivation and algorithm description.
- The numerical results look promising
- The design of DSGD-AAU looks pretty novel to me.

Weaknesses:
- Since the DSGD-AAU is designed to mitigate the stragglers issue without suffering from the gradient staleness, how are these improvements reflected in the convergence result? Which parameters depend on the gradient staleness, and how DSGD-AAU improves (or provides a trade-off) them? I assume that it is related to the Connectivity Time $B$ in Assumption 4.2. It would be interesting to investigate the theoretical improvement (or trade-off) of the AAU design.
- It seems that the analysis of the main theorem is independent of the path search procedure? I am hoping for more fine-grained analysis which would explain the numerical improvement.
- I am not sure about how Algorithm 2 implements the idea of selecting a subset of the fastest computational workers $\mathcal{N}(k)$. Could the authors elaborate more?

---

> ### Author Response · Authors · 2023-08-03
> **Response to Reviewer Tbdh (1/3)**
>
> **Weakness #1:** Since the DSGD-AAU is designed to mitigate the stragglers issue without suffering from the gradient staleness, how are these improvements reflected in the convergence result? Which parameters depend on the gradient staleness, and how DSGD-AAU improves (or provides a trade-off) them? I assume that it is related to the Connectivity Time $B$  in Assumption 4.2. It would be interesting to investigate the theoretical improvement (or trade-off) of the AAU design.
>
> **Our Response:** Thank you for your insightful comments. As shown in the examples in Figure 1b and Figure 2, our proposed DSGD-AAU can indeed reduce the impact of stale gradient information. From a theoretical perspective, since we do not make any staleness assumption, you are right that parameter $B$ largely depends on the gradient staleness. As illustrated in the Pathsearch (Algorithm 3) procedure, which is a key part in the DSGD-AAU algorithm, we can guarantee that parameter $B$ is upper-bounded by $N-1$, which is however cannot be bounded in conventional asynchronous DSGD algorithms (e.g., Lian et al. 2018, Assran Rabbat 2020, Yang et al. 2021).  Basically, our algorithm tries to reduce the value of $B$ as much as we can. According to Theorem 4.6 and Corollary 4.8, we have the following inequality:
> \begin{align*}
>     \sqrt{\frac{N}{K}}\leq \sqrt{\frac{(1-q)^2}{30C^2L^2N}+\frac{9N^4}{16}}-\frac{3N^2}{4}\leq \sqrt{\frac{(1-q)^2}{30C^2L^2N}}\leq \sqrt{\frac{1}{30C^2L^2N}} ,
> \end{align*}
> which leads to the fact that $K\geq 30C^2L^2N^2.$
> The benefit of reducing the value of $B$ can be found in the condition that $K\geq 30C^2L^2N^2$. As we can observe and evaluate that the value of $C$ is monotonically increasing as the value of $B$ increases. Therefore, the required iteration number $K$ will become larger when $B$ increases. Hence, reducing the value of $B$ can lower the required number of iterations.  This key observation motivates our design of DSGD-AAU in Section 5, which aims to jointly dynamically select the active workers per round and minimize the connectivity time $B$, which resolves the staleness issue in conventional asynchronous algorithms (Luo et al., 2020, Assran et. al., 2020, Wang et. al., 2022). However, we admit that theoretically characterizing how much benefit it gains requires more assumptions on the topology and gradient staleness, which we leave for future work.
>
> We added the following discussions at the end of Section 4 to make it more intuitive: "An interesting observation is that the value of $C > 1$ is monotonically increasing as $B$ increases. Therefore, the required iteration number $K$ will become larger if the bounded-connectivity time $B$ increases. Hence, reducing the value of $B$ can lower the required number of iterations $K$. This key observation motivates our design of DSGD-AAU in Section 5, which aims to jointly dynamically determine the fastest workers set per round and minimize the connectivity time $B$. This resolves the staleness issue in conventional asynchronous algorithms Luo et al.,2020); Assran \& Rabbat, 2020; Wang et al., 2022."
>
>
> **Weakness #2:** It seems that the analysis of the main theorem is independent of the path search procedure? I am hoping
>
> **Our Response:** Please kindly refer to our above response to **Weakness #1**. Since reducing $B$ can actually benefit the training process, our goal is to find a way to reduce $B$ as much as we can. The Pathsearch procedure can guarantee that $B$ will be upper bounded by $N-1$. So it is fully motivated by our theoretical results.

---

> > ### Author Response · Authors · 2023-08-03
> > **Response to Reviewer Tbdh (2/3)**
> >
> > **Weakness #3:** I am not sure about how Algorithm 2 implements the idea of selecting a subset of the fastest computational workers N(k). Could the authors elaborate more?**
> >
> > **Our Response:** In Algorithm 2, the selection of a subset of the fastest computational workers, denoted as $\mathcal{N}(k)$, is performed dynamically at each iteration. This subset is determined based on the workers that finish their local computations the fastest. Here's a more detailed explanation:
> >
> > 1. All workers start their local computations simultaneously.
> >
> > 2. The worker that finishes its local computation first, denoted as $j_k$, waits until any of its neighboring workers finish their computations.
> >
> > 3. If two other workers, say $i$ and $j$, successfully exchange their local updates and the edge $(i, j)$ is not already in the set of established edges $\mathcal{P}$, then this edge and the nodes $i$ and $j$ are added to $\mathcal{P}$ and $\mathcal{V}$, respectively.
> >
> > 4. If the established edge is between two nodes that are already in $\mathcal{V}$, then it is not added to $\mathcal{P}$, and the current iteration continues until an edge is established with at least one node not in $\mathcal{V}$.
> >
> > 5. This process is repeated for each iteration, dynamically determining the subset of the fastest workers $\mathcal{N}(k)$.
> >
> > An illustrative example of DSGD-AAU for 4 heterogeneous workers with a fully-connected network topology is as follows. At the 1st iteration ($k=1$), worker $4$ first completes the local computation, it waits until the neighbor worker $1$ finishes the local computation. Then, workers $4$ and $1$ exchange the updates and  DSGD-AAU stores nodes $1, 4$ into $\mathcal{V}$, and edge $(1,4)$ into $\mathcal{P}$. This terminates the 1st iteration. At the 3rd iteration ($k=3$), worker $4$ first completes the local computation, it waits until worker $1$ completes the local computation.  However, since nodes $1,4$ have been stored in $\mathcal{V}$ and edge $(1,4)\in\mathcal{P}$, worker $4$ keeps waiting until worker $2$ completes the local computation.  Workers $1,2,4$ exchange updated information, and DSGD-AAU stores new worker $2$ in $\mathcal{V}$ and $\mathcal{P}=\{(1,4), (2,3), (1,2), (2,4)\}$, which ends the 3rd iteration.  Since there exists a path for any arbitrary two workers in $\mathcal{G}^\prime:=\{\mathcal{P},\mathcal{V}\}$ with $\mathcal{V}=\mathcal{N}$, DSGD-AAU resets $\mathcal{V}$ and $\mathcal{P}$ as empty sets and move to the next iteration.
> >
> > This procedure allows the algorithm to adapt to the heterogeneity in computational speeds among the workers, thus addressing the straggler issue in distributed SGD. It also reduces the communication overhead as each worker does not need to communicate with all other workers, but only with a subset of them.

---

> > > ### Author Response · Authors · 2023-08-03
> > > **Response to Reviewer Tbdh (3/3)**
> > >
> > > **Our Continued Response to Weakness #3**: To reflect the change, we added a discussion in Section 5 as:
> > > **Pathsearch.** At each iteration $k$, we denote the worker which first completes the local parameter update as $j_k$. Worker $j_k$ keeps idle until any one of the following two situations occurs. The first case is that one of worker $j_k$'s neighbor $i\in\mathcal{N}_{j_k}$ completes the local parameter update. If edge $(j_k,i)\in\mathcal{E}\cap(j_k,i)\notin \mathcal{P}$ and worker $j_k\notin\mathcal{V} \bigcup i\notin\mathcal{V}$, then store edge $(j_k,i)$ into $\mathcal{P}$ and nodes $\{i, j_k\}$ into $\mathcal{V}$. Once this is achieved, all workers move into the $(k+1)$-th iteration. The other case is that any other new edge $(i_1, j_1)$ between workers $i_1$ and $j_1$ other than $j_k$ is established such that edge $(i_1,j_1)\in\mathcal{E}\cap(i_1,j_1)\notin \mathcal{P}$ and node $j_1\notin\mathcal{V} \bigcup i_1\notin\mathcal{V}$. Then all workers move to the  $(k+1)$-th iteration. Intuitively, for both cases we guarantee that there is at least one new worker (with new updated parameters) involved in each iteration, no matter the new worker is a neighbor or for of worker $j_k$. This process is repeated for each iteration, which dynamically determines the subset of the fastest workers $\mathcal{N}(k)$. A detailed description of Pathsearch is provided in Algorithm 3 in ``Section B. The Proposed Path Searching Procedure'' in the appendices.
> > > An illustrative example of DSGD-AAU for 4 heterogeneous workers with a fully-connected network topology is presented in Figure 2. At the 1st iteration ($k=1$), worker $4$ first completes the local computation, it waits until the neighbor worker $1$ finishes the local computation. Then, workers $4$ and $1$ exchange the updates and  DSGD-AAU stores nodes $1, 4$ into $\mathcal{V}$, and edge $(1,4)$ into $\mathcal{P}$. This terminates the 1st iteration. At the 3rd iteration ($k=3$), worker $4$ first completes the local computation, it waits until worker $1$ completes the local computation.  However, since nodes $1,4$ have been stored in $\mathcal{V}$ and edge $(1,4)\in\mathcal{P}$, worker $4$ keeps waiting until worker $2$ completes the local computation.  Workers $1,2,4$ exchange updated information, and DSGD-AAU stores new worker $2$ in $\mathcal{V}$ and $\mathcal{P}=\{(1,4), (2,3), (1,2), (2,4)\}$, which ends the 3rd iteration.  Since there exists a path for any arbitrary two workers in $\mathcal{G}^\prime:=\{\mathcal{P},\mathcal{V}\}$ with $\mathcal{V}=\mathcal{N}$, DSGD-AAU resets $\mathcal{V}$ and $\mathcal{P}$ as empty sets and move to the next iteration.

---

### Review · Reviewer_6FCT · 2023-07-10

**Summary Of Contributions:**

This article introduces a decentralized SGD algorithm to address the staleness and heavy communication issue of existing fully asynchronous SGD methods. An adaptive asynchronous update is proposed and  both theoretical and numerical results of this algorithm are provided .

**Audience:**

Yes

**Broader Impact Concerns:**

The article proposes a fully-decentralized algorithm to achieve a balance between synchronous  and asynchronous SGD methods. I think the impact is still not so clear from the numerical results. A major revision is recommended .

**Claims And Evidence:**

No

**Requested Changes:**

- clarify what is non i.i.d dataset in the paper / appendix .
- make the remark 4.7 more clear as when eta -> 0, 1/eta -> infinity, so it is not clear whether the bound in eq 7 is small.
- make the corollary 4.8 more clear, in terms of the condition of the learning rate eta. Should eta be smaller than 1/L, as in Theorem 4.6?
- clarify how it works the step 6,7,8 in the Algorithm 2, it is not clear whether you need to use a lock or not in the fully decentralized scenario.
- add a large dataset in your numerical results to show the interest of proposed method in large-scale training .
- vary the evaluation accuracy level to measure the speedup of the method as I think 65% is too low. I am not fully convinced that the method is going to have a linear speedup as the theoretical results is only about the ergodic average solution .
- the learning rate scheduling with eta(k) = eta0 * delta^k seems to be decrease too quickly, why do you choose it in this way ?
- Report the communication time as well to support the results in Fig 5?
- Add a discussion on what would happen to the i.i.d. dataset case?


**Strengths And Weaknesses:**

The strength of this article is to introduce a novel idea to achieve a good balance between synchronous  and asynchronous SGD methods . The article is well written .

I think the Algorithm 2 should be further clarified in terms of how the synchronization at step 8 is performed (if needed). The main weakness is the evaluation of the idea. In numerical results, only CIFAR dataset results are provided, and the linear-speedup is not so clear as the it is measured at a fixed (relatively small) accuracy level.

---

> ### Author Response · Authors · 2023-08-03
> **Response to Reviewer 6FCT (1/5)**
>
> Thank you for your constructive feedback and comments. Here we would like to address the reviewer's concerns and our point-to-point responses are as follows:
>
> **Weakness #1:** I think the Algorithm 2 should be further clarified in terms of how the synchronization at step 8 is performed (if needed).
>
> **Our Response:** Thank you for this comment and providing us an opportunity for further clarifying Algorithm 2. Firstly, we need to emphasize that our proposed DSGD-AAU is an adaptive asynchronous algorithm as the workers update their parameters at different paces. This is fundamentally different from synchronous learning algorithms where all workers update their parameters at the same pace (see Figure 1a for an illustration). The synchronization in Algorithm 2 at Step 8 refers to the process of updating the model parameters based on the computations performed by the workers. This is done through a so-called gossip-average process, where the worker completes its local gradient computation first and then waits for its neighbor workers to complete their computations. This is a common practice in decentralizing learning/training literature. Specifically, in Step 8, the synchronization occurs across workers which have completed the gradient computation and local gradient descent step. Those workers broadcast the up-to-date parameters to their neighbors and receive the updated parameters from their neighbors which have completed the local gradient computation. Parameter sharing only occurs between neighborhood workers. Different from existing works with synchronous updates that suffer from stragglers and asynchronous updates that suffer from staleness, we proposed an adaptive asynchronous update method to address these issues, which provide an input to the update in Step 8.
>
> With that being said, Algorithm 2 goes as follows. At each iteration $k$, worker $j_k$ performs local SGD (lines 2-4) followed by one step of the graph searching procedure to establish a new edge in set P (line 5). The next operation in the current iteration is the gossip average of the local parameters of workers $j_k$ in the set $\mathcal{N}\_{j_k}(k)$ (line 6). Note that if any other worker $i_k$ completes the local gradient update during the Pathsearch of the fast worker $j_k$, it will also run Pathsearch independently until $\mathcal{P}$ and $\mathcal{V}$ get updated. Hence, worker $i_k$ also conducts gossip-average of the local parameters in the set $\mathcal{N}_{i_k}(k)$ (lines 7-9).
>
> To bring our DSGD-AAU into practice, we implement it in PyTorch on Python 3 with three NVIDIA RTX A6000 GPUs using the Network File System (NFS) and MPI backends, where files are shared among workers through NFS and communications between workers are via MPI. Note that both NSF and MPI have been widely used in the implementation of decentralized algorithms, where MPI is utilized to enable parallel execution of all workers and facilitate inter-worker communication.
>
> **Weakness #2:** The main weakness is the evaluation of the idea. In numerical results, only CIFAR dataset results are provided, and the linear-speedup is not so clear as the it is measured at a fixed (relatively small) accuracy level.
>
> **Our Response:** We are afraid that there is a misunderstanding about our evaluation in Section 6.
>
> First, as stated in Section 6, "**Dataset and Model**", we considered **two tasks** (image classification and natural language processing) using **four datasets**. Specifically, for image classification tasks, we use CIFAR-10, MNIST and Tiny-ImageNet datasets, and for NLP task, we use the Shakespeare dataset. For ease of readability, we only present the results on CIFAR-10 in the main paper, and relegate the corresponding results on other datasets to the appendices (see "**Section D. Additional Experimental Results**") since we draw similar conclusions from these datasets as from CIFAR-10.
>
> Second, as stated in **"Speedup and Communication"** in Section 6, we compute the speedup of different algorithms with respect to the DSGD with full worker updates as in state-of-the-art methods AD-PSDG (Lian et al. 2018) and Prague (Luo et al. 2020). In Figure 5, we report results using ResNet-18 on non-i.i.d. CIFAR-10, for which the highest test accuracy that "DSGD with full worker updates" can achieve is a little above 65\% (note that we consider non-i.i.d. settings).  Since "DSGD with full worker updates" is the baseline to compute the speedup for all algorithms we compare, including AGP, AD-PSGD, Prague and DSGD-AAU, we choose 65\% as the target test accuracy. Note that similar observations can be made for other models and datasets (see the appendices, i.e., Figures 6, 7, 8 for other models on CIFAR-10, and our new results using other datasets and models in Figures 9-14), i.e., DSGD-AAU is robust across different models and datasets to achieve the fastest convergence time and the best speedup while maintaining comparable or even better communication efficiency.

---

> > ### Author Response · Authors · 2023-08-03
> > **Response to Reviewer 6FCT (2/5)**
> >
> > **Requested Change #1:** Clarify what is non i.i.d dataset in the paper/appendix.
> >
> > **Our Response:** We actually provided a detailed discussion on how to generate non-i.i.d. dataset in the appendices (see "**Section D. Additional Experimental Results**" under "Dataset and Model"). Thank you for this question. To make it clear and for ease of readability, we provide an additional discussion in Section 6 and point interested readers to our appendices for further discussions.
> >
> > **Requested Change #2:** Make the remark 4.7 more clear as when eta -> 0, 1/eta -> infinity, so it is not clear whether the bound in eq 7 is small.
> >
> > **Our Response:** We are afraid that there is a misunderstanding.  If $\eta$ goes to 0, the bound in Eq. (7) is meaningless and $K$ needs to be very large.  The reason that we cannot make $\eta$ go to zero is that $\eta$ is the learning rate, and it does not make any sense to consider a training process with a zero learning rate. To the best of our knowledge, there is no paper in the literature that consider a zero-learning rate (i.e., $\eta\rightarrow 0)$, and all existing bounds in the literature are in the case that $\eta$ is meaningful, i.e., there often exists a $\frac{1}{\eta}$ term in the bounds [Lian et al. 2018][Yang et al. 2021]. More importantly, in Corollary 4.8, we provide an explicit value of $\eta$, i.e., $\eta=\sqrt{N/K}$ so as to achieve the linear speedup in convergence.
> >
> > **Requested Change #3:** Make the corollary 4.8 more clear, in terms of the condition of the learning rate  eta. Should eta be smaller than 1/L, as in Theorem 4.6?
> >
> > **Our Response:** Thank you for this great question and sharp observation. To be clear, the condition on $K$ should be  $K\geq \max\left(\frac{2N}{\frac{(1-q)^2}{30C^2L^2N}+\frac{9N^4}{16}-\frac{\sqrt{3}N^{3/2}(1-q)}{CL\sqrt{40}}}, NL^2\right)$. This directly comes from the condition in Theorem 4.6 and Corollary 4.8 that
> > \begin{align*}
> >     \sqrt{\frac{N}{K}}\leq \min\left(\sqrt{\frac{(1-q)^2}{30C^2L^2N}+\frac{9N^4}{16}}-\frac{3N^2}{4}, 1/L\right).
> > \end{align*}
> > Since the following inequality holds,
> > \begin{align*}
> >     \sqrt{\frac{(1-q)^2}{30C^2L^2N}+\frac{9N^4}{16}}-\frac{3N^2}{4}\leq \sqrt{\frac{(1-q)^2}{30C^2L^2N}}\leq \sqrt{\frac{1}{30C^2L^2N}},
> > \end{align*}
> > we can  simplify the condition on $K$ as $K\geq \max(30C^2L^2N^2, NL^2).$ Since $C>1, N>1$, it always holds that $30C^2L^2N^2>NL^2$. Thus, we have the final condition on $K$ as $K\geq 30C^2L^2N^2.$ We have updated Corollary 4.8 accordingly.
> >
> > **Requested Change #4:** Clarify how it works the step 6,7,8 in the Algorithm 2, it is not clear whether you need to use a lock or not in the fully decentralized scenario.
> >
> > **Our Response:** Thank you for this comment and providing us an opportunity for further clarifying Algorithm 2. As we responded earlier to your **Weakness #1**, steps 6-8 in Algorithm 2 work as follows for each iteration $k$:
> >
> > 1. Each worker $j_k$  computes its local update $\tilde{\mathbf{w}}_{j_k}(k)$ using its local dataset $D_i$. This is done by computing the gradient of the loss function with respect to the current model parameters $\mathbf{w}_i(k)$. The local update is then the negative of this gradient, scaled by the learning rate $\eta$.
> >
> > 2. Each worker $j_k$ then sends its local update $\tilde{\mathbf{w}}\_{j_k}(k)$ to its neighbor workers in the set $\mathcal{N}\_{j_k}(k)$. This is done in a fully decentralized manner, meaning that there is no central server or coordinator involved in this process. Instead, each worker communicates directly with all other neighbor workers in the set $\mathcal{N}_{j_k}(k)$.
> >
> > 3. Each worker $j_k$ then updates its model parameters $\mathbf{w}_{j_k}(k+1)$ by adding the average of all received local updates from neighbors to its current model parameters. This is done using the gossip-average framework.
> >
> > Regarding your question about the need for a lock in a fully decentralized scenario, this paper does not need the use of locks in these steps.  This is due to the fact that we propose an adaptive asynchronous update scheme that achieves the best of both worlds of the conventional synchronous and fully asynchronous update schemes.   In our proposed DSGD-AAU, each worker computes its local update, sends it to all other neighbor workers, and updates its model parameters independently of all other workers. This is one of the main advantages of the proposed DGSD-AAU as it can operate efficiently and robustly in large-scale, distributed settings without the need for central coordination or synchronization.

---

> > > ### Author Response · Authors · 2023-08-03
> > > **Response to Reviewer 6FCT (3/5)**
> > >
> > > **Requested Change #5:** Add a large dataset in your numerical results to show the interest of proposed method in large-scale training.
> > >
> > > **Our Response:** Thank you for your question. As stated in Section 6, "**Dataset and Model**", we have considered **two tasks** (image classification and natural language processing) using **four datasets**.  Specifically, for image classification tasks, we use three datasets: CIFAR-10, MNIST and Tiny-ImageNet, and for NLP task, we use the Shakespeare dataset.  Note that both Tiny-ImageNet and Shakespeare datasets are regarded as large datasets in decentralized or federated learning literature.  In particular, the Shakespeare dataset consists of 3,564,579 characters in the training set and 870,014 characters in the test set. It is important to note that this dataset exhibits significant class imbalance, with a large number of roles having only a few lines while a few roles have a substantial number of lines, which is a good fit for large-scale non-i.i.d. settings.  Please kindly refer to the additional discussions on these datasets ``Section D. Additional Experimental Results'' in the appendices.
> > >
> > > **Requested Change #6:** Vary the evaluation accuracy level to measure the speedup of the method as I think $65\%$ is too low. I am not fully convinced that the method is going to have a linear speedup as the theoretical results is only about the ergodic average solution.
> > >
> > > **Our Response:** Thank you for this question. Please kindly refer to our response to **Weakness #2.** In Figure 5, we report the results using ResNet-18 on non-i.i.d. CIFAR-10, for which the highest test accuracy that "DSGD with full worker updates" can achieve is a little above 65\% (note that we consider non-i.i.d. settings). Since "DSGD with full worker updates" is the baseline to compute the speedup for all algorithms we compare, including AGP, AD-PSGD, Prague and our DSGD-AAU, we choose the 65\% as the target test accuracy. Note that our speedup results are consistent across all datasets and models, rather than just the case using ResNet-18 using non-i.i.d. CIFAR-10 shown in Figure 5. These can be observed in Figures 6, 7, 8, and 9-14 in the appendices. We added a "Footnote 3" at the end of Section 6 to further clarify this.
> > >
> > > **Requested Change #7:** The learning rate scheduling with $\eta(k) = \eta_0 * \delta^k$ seems to be decrease too quickly, why do you choose it in this way ?
> > >
> > > **Our Response:** We utilized a decaying learning rate, i.e., $\eta(k)=\eta_0\cdot\delta^k$ in our experiments. This is a practice in existing works (including but not limit to the ones listed below) to speed up the convergence process in experiments via leveraging decaying learning rates. For example, the decay ratio $\delta$ has been widely set between $0.94$ and $0.99$ in aforementioned state-of-the-art works, and we set it to be $0.95$ in our experiments.
> > >
> > > [Chen et al. 2016] Jianmin Chen, Xinghao Pan, Rajat Monga, Samy Bengio, and Rafal Jozefowicz. Revisiting Dis- tributed Synchronous SGD. arXiv preprint arXiv:1604.00981, 2016.
> > >
> > > [Le et al. 2015] Ya Le and Xuan Yang. Tiny ImageNet Visual Recognition Challenge. CS 231N, 7:7, 2015.
> > >
> > > [LeCun et al. 1988] Yann LeCun, Léon Bottou, Yoshua Bengio, and Patrick Haffner. Gradient-Based Learning Applied to Document Recognition. Proceedings of the IEEE, 86(11):2278–2324, 1998.
> > >
> > > [Lian et al. 2018] Xiangru Lian, Wei Zhang, Ce Zhang, and Ji Liu. Asynchronous Decentralized Parallel Stochastic Gradient Descent. In Proc. of ICML, 2018.
> > >
> > > [Luo et al. 2020] Qinyi Luo, Jiaao He, Youwei Zhuo, and Xuehai Qian. Prague: High-Performance Heterogeneity- Aware Asynchronous Decentralized Training. In Proc. of ACM ASPLOS, 2020.
> > >
> > > [Ruan et al. 2020] Yichen Ruan, Xiaoxi Zhang, Shu-Che Liang, and Carlee Joe-Wong. Towards Flexible Device Participation in Federated Learning for Non-IID Data. arXiv preprint arXiv:2006.06954, 2020.
> > >
> > > [Simonyan et al. 2015] Karen Simonyan and Andrew Zisserman. Very Deep Convolutional Networks for Large-scale Image Recognition. In Proc. of ICLR, 2015.
> > >
> > > [Tang et al. 2020] Zhenheng Tang, Shaohuai Shi, Xiaowen Chu, Wei Wang, and Bo Li. Communication-Efficient Distributed Deep Learning: A Comprehensive Survey. arXiv preprint arXiv:2003.06307, 2020.
> > >
> > > [Wang et al. 2022] Haoxiang Wang, Zhanhong Jiang, Chao Liu, Soumik Sarkar, Dongxiang Jiang, and Young M Lee. Asynchronous training schemes in distributed learning with time delay. arXiv preprint arXiv:2208.13154, 2022.
> > >
> > > [Yang et al. 2021] Haibo Yang, Minghong Fang, and Jia Liu. Achieving Linear Speedup with Partial Worker Participa- tion in Non-IID Federated Learning. In Proc. of ICLR, 2021.
> > >
> > > [Zhao et al. 2018] Yue Zhao, Meng Li, Liangzhen Lai, Naveen Suda, Damon Civin, and Vikas Chandra. Federated Learning with Non-IID Data. arXiv preprint arXiv:1806.00582, 2018.

---

> > > > ### Author Response · Authors · 2023-08-03
> > > > **Response to Reviewer 6FCT (4/5)**
> > > >
> > > > **Requested Change #8:** Report the communication time as well to support the results in Fig 5?
> > > >
> > > > **Our Response:** Thank you for this great question, and again we are afraid that there are discussions provided in the paper that may be missed.  In particular, we provided a discussion in **"Section C.4. Assumption on Adaptive Asynchronous Update"** in the appendices.  We provide an extended discussion here.
> > > >
> > > > Communication delays are much smaller and dominated by computational delays in our considered settings. This is due to the fact that in this paper, we focused on stragglers resulting from computational limits, i.e., workers with slow computational capabilities. We would like to mention that incorporating both communication and computation stragglers will significantly complicate the convergence analysis, since the parameter update delays caused by these two types of stragglers contribute to the final convergence in a totally different manner. This unified convergence has not been characterized yet. To the best of our knowledge, there is no such theoretical study in the literature either. In this paper, we first proved that DSGD-AAU achieves a linear speedup for convergence in terms of iterations. In such case, the communication does not affect the result with respect to the number of iterations (computation). However, the communication time will impact the wall-clock time for convergence.
> > > >
> > > > To the best of knowledge, Prague (Luo et al., 2020) is perhaps the first work to separately measure the computation time and communication time by running different number of workers and found that the communication time is dominated by the computation time in the settings similar to ours. Unfortunately, there is no convergence analysis in Prague. Moreover, our convergence time in the numerical evaluation considers the total time of the whole process, which includes both computation and communication time.
> > > > We further explore this and estimate the communication time in our system given the network speed. For instance, in our system, the network speed is about 20~GB/s given the hardware of our server (memory card), and **the communication time only accounts for 0.14\%-4\% of the total time measured in communication and computation time under the CIFAR-10 dataset.** This is **in alignment with our model and motivations that we focus on computational stragglers in this paper.** Exactly same thing has been done in Prague (Luo et al., 2020), AD-PSGD (Lian et al., 2018), and Li et al. (Li et al., 2020a;b). Nevertheless, our proposed DSGD-AAU algorithm can still work if communication stragglers exist. Moreover, to further validate performance of DSGD-AAU from the communication's perspective, we provide experimental results in Figure 5 (Section 6) and Figures 6-14 (Appendix D). We observe that DSGD-AAU consistently
> > > > outperforms baseline methods and achieves the best speedup at no cost of additional communication.
> > > >
> > > > For ease of readability and to further clarify our settings for report the results in Figure 5, we add the following arguments in Section 6: "Need to mention that  our convergence time in the numerical evaluation considers the total time of the whole process, which includes both computation and communication time. since we mainly focus on computational stragglers in this paper, the communication time are dominated by computational time in our considered setting. A more detailed discussion can be found in Section C.4 in the appendices".

---

> > > > > ### Author Response · Authors · 2023-08-03
> > > > > **Response to Reviewer 6FCT (5/5)**
> > > > >
> > > > > **Requested Change #9:** Add a discussion on what would happen to the i.i.d. dataset case?
> > > > >
> > > > > **Our Response:** Once again, we are afraid that there is a misunderstanding here.  In this paper, both i.i.d. and non-i.i.d. cases are considered. First, in our theoretical analysis, we are mainly motivated with real-world scenarios with non-i.i.d. datasets among workers. In particular,  the heterogeneity of non-i.i.d. datasets among different workers is quantified via a universal bound $\varsigma$ (Assumption 4.5). This general framework capture the i.i.d. dataset case, i.e., the  i.i.d. datasets refer to a special case where $\varsigma = 0$.  This is reflected in the convergence bound (Theorem 4.6) with a non-vanishing constant term $\frac{2\eta}{N}\varsigma^2.$
> > > > >
> > > > > Second, in our experimental evaluations, we also consider both non-i.i.d. and i.i.d. cases. Again, since we focus on non-i.i.d. cases, we present most of the experimental results for non-i.i.d. cases, especially in the main paper, and relegate the results on i.i.d. cases to Section D in the appendices. Additional Experimental Results (e.g., Tables 10 and 11).  Specially, for the i.i.d. case, this paper presents the test accuracy of different models using different i.i.d. datasets with 128 workers in Table 10 of the paper. We also summarize the test accuracy of different models over different i.i.d. datasets when trained for a limited time in Table 11 with different numbers of workers.These results show that DSGD-AAU consistently achieves a higher test accuracy compared to other methods, both for i.i.d. and non-i.i.d. datasets.
> > > > >
> > > > > To further clarify these results and based on the reviewer's suggestion, we modified our discussions on  results on i.i.d. datasets in Section D as "Similar observations can be made when considering the i.i.d. CIFAR-10, which is shown in Figures 11 and 12,
> > > > > as well as other models and datasets. In particular, the heterogeneity of non-i.i.d. datasets among different
> > > > > workers is quantified via a universal bound $\varsigma$ (Assumption 4.5). This general framework captures the i.i.d.
> > > > > dataset case, i.e., the i.i.d. datasets refer to a special case where $\varsigma = 0$. This is reflected in the convergence
> > > > > bound (Theorem 4.6) with a non-vanishing constant term $\frac{2\eta}{N}\varsigma^2.$ Especially, for the i.i.d. case, we present
> > > > > the test accuracy of different models using different i.i.d. datasets with 128 workers in Table 10. We also
> > > > > summarize the test accuracy of different models over different i.i.d. datasets when trained for a limited time
> > > > > in Table 11 with different numbers of workers. These results show that DSGD-AAU consistently achieves a
> > > > > higher test accuracy compared to other methods, both for i.i.d. and non-i.i.d. datasets."

---

> > > > > > ### Comment · Reviewer_6FCT · 2023-08-21
> > > > > > **Further clarifications and evaluations**
> > > > > >
> > > > > > Dear authors,
> > > > > >
> > > > > > Thanks for providing detailed answers to my questions. I have read the revised version of the article and have some further questions regarding your results. I think your evaluations is still limited in some sense.
> > > > > >
> > > > > > - Clarify the test accuracy achieved by the DSGD  with full worker updates: what this name means in terms of the algorithm? Could you provide a reference of this algorithm which achieves the test accuracy (around) 65%? As you are not comparing this method directly in Fig 5, I would suggest to have discussion on the speedup to achieve a higher accuracy, e.g. 73%. Provide an explanation of why the  DSGD  with full worker updates could not achieve this accuracy level (if possible).
> > > > > >
> > > > > > - Clarify the definition of the iteration in x-axis Fig 3 for each method. If you are using a different notion for each algorithm, then I would add a plot with wallclock time to make these methods comparable. Also are you evaluating the performance of the average 1/N sum_i w_i(k) at iteration k for each method ? Could you mention how do you obtain 1/N sum_j w_j (k) in the decentralized and asynchronous setting ?
> > > > > >
> > > > > > - Topology of the networks: as you mentioned the deadlock issue of AD-PSGD, are you considering only the bipartite topology in your simulations to compare with this method? Could you explain why it is possible to achieve a 60 speedup with only 3 GPUs in your Fig 5 ? Does this result carry over to larger datasets ? How about to use other topology of the commutation networks?
> > > > > >
> > > > > > - Name of the algorithms: I think it is a bit confusing to call Algo 1 in this paper an asynchronous method. Could you add some reasoning about the choice of this name ? As you are using a global counter k to make updates of each w_j. I would call it a synchronous method.

---

> > > > > > > ### Author Response · Authors · 2023-08-24
> > > > > > > **Response to follow-up questions (1/3)**
> > > > > > >
> > > > > > > We thank the reviewer for your valuable time on evaluating our work.  In the following, we would like to address the reviewer's concerns and present our point-to-point responses.
> > > > > > >
> > > > > > > **Response to Question 1:** When evaluating the speedup of our DSGD-AAU in terms of convergence in wall-clock time, it is important to show how the speedup scales with the number of workers when compared with the centralized training where there is only one worker. However, since we mainly focus on addressing the stragglers issues in decentralized training in this paper, to maintain a fair comparison (i.e., we need to measure the wall-clock time of each iteration in centralized training with one worker), the term "DSGD with Full Worker Updates"  refers to a baseline algorithm that a central server waits for all workers to update their parameters in every communication round, which is similar as FedAvg algorithm with computation delay due to stragglers.
> > > > > > >  For the non-i.i.d. CIFAR-10 dataset, the highest test accuracy that "DSGD with full worker updates" can achieve is slightly above 65\% when the number of workers is 32. The lower test accuracy is widely observed for the non-i.i.d. settings.  For example, [Yang et al. 2021] has also investigated the impact of non-i.i.d on the training loss and test accuracy.   As the degree of non-i.i.d. datasets increases, its negative impact on the convergence is
> > > > > > > becoming more obvious. The higher the non-i.i.d. degree, the slower the convergence speed. As
> > > > > > > the non-i.i.d. degree increases, it is obvious that the training loss is
> > > > > > > increasing and the test accuracy is decreasing. For those with a higher degree of non-i.i.d., the convergence
> > > > > > > curves oscillate and are highly unstable. This phenomenon can be clearly observed in Figures 5-6 on page 22 in [Yang et al. 2021].
> > > > > > > As we can see from Figures 5-6, the test accuracy for CIFAR-10 can be even below 60\% under non-i.i.d. datasets.
> > > > > > >
> > > > > > > [Yang et al. 2021] Achieving Linear Speedup with Partial Worker Participation in Non-IID Federated Learning, ICLR 2021.
> > > > > > >
> > > > > > >
> > > > > > > **Response to Question 2:** The iteration in x-axis of Figure 3 represents the number of communication rounds. For centralized settings, one iteration corresponds to one round of communication between the parameter server and the workers. For decentralized settings, one iteration corresponds to one round of gossip among workers. For the asynchronous setting as considered in this paper, one iteration corresponds to one round of communication between the subset of the fastest computational workers.
> > > > > > >  As we presented in Algorithm 1 and Section 3, at each iteration $k$, the key is to determine the set of fastest workers, i.e., $\mathcal{N}(k)\in\mathcal{N}$,  which completed the local gradient computations, and only workers in set $
> > > > > > > \mathcal{N}(k)$ actively conduct gossip-average with neighbors while workers in set $\mathcal{N}\setminus\mathcal{N}(k)$ are still computing their local gradients. The iteration $k$ begins at finding the set $\mathcal{N}(k)$ and ends when $\mathcal{N}(k)$ is determined with workers inside it having completed the local gradient computations and the gossip-average operations.
> > > > > > >
> > > > > > >  The convergence in wall-clock time of ResNet-18 on non-i.i.d. CIFAR-10
> > > > > > > dataset with 128 workers is presented in Figure 4. Again, we observe that DSGD-AAU converges faster,
> > > > > > > and importantly, DSGD-AAU achieves a higher accuracy for a given training time. We even evaluated the linear speedup in terms of wall-clock time in Figure 5(a) and  Figures 6(a)-14(a) in the appendices.
> > > > > > >
> > > > > > > For the evaluation process, AD-PSGD utilizes the average of models from all workers for inference, and we have also adopted exactly the same approach in state-of-the-art asynchronous methods, e.g., AD-PSGD. Specifically, we designate a worker as the evaluation worker. In addition to conducting local training and communicating with its neighboring workers, this evaluation worker gathers locally trained models at each iteration. Once the collection is complete, the worker uses stored global test data samples to evaluate the global training loss. We have implemented this methodology not only for our evaluation but also to assess other state-of-the-art approaches. This consistent evaluation approach enables a comprehensive assessment of various techniques.

---

> > > > > > > ### Author Response · Authors · 2023-08-24
> > > > > > > **Response to follow-up questions (2/3)**
> > > > > > >
> > > > > > > **Response to Question 3:** Thank you for this comment and providing us an opportunity for further clarifying our experiments.
> > > > > > > First, we need to emphasize that our work considers randomly generated topologies, which contain a broad class of topologies as long as satisfying the strongly-connected conditions (Assumption 4.2). As you know, a deadlock arises when multiple communications wait for each other in a cyclic manner in the benchmark algorithm AD-PSGD. Specifically, when a communication is in need of a resource locked by another communication, it remains in a waiting state until either the lock is granted or a timeout period elapses. To address this issue, we leverage the timeout method, which determines the duration of a communication that is willing to wait before abandoning its attempt to acquire the resource. The timeout method has been extensively utilized in FedScale [Lai et al. 2022] and other related works [Li et al. 2022a, Li et al. 2022b, Qu et al. 2022, Lai et al. 2021]  to prevent prolonged and unnecessary waiting.
> > > > > > >
> > > > > > >
> > > > > > > Second,  each GPU accommodates approximately 43 workers, and these workers are able to utilize the same GPU for local training, as the available memory space is sufficient to accommodate their operations.
> > > > > > >
> > > > > > > Third, as stated in Section 6, **"Dataset and Model"**, we have considered **two tasks**  (image classification and natural language processing) using **four datasets** .  Specifically, for image classification tasks, we use three datasets: CIFAR-10, MNIST and Tiny-ImageNet, and for NLP task, we use the Shakespeare dataset.  Note that both Tiny-ImageNet and Shakespeare datasets are regarded as large datasets in decentralized or federated learning literature.  In particular, the Shakespeare dataset consists of 3,564,579 characters in the training set and 870,014 characters in the test set. It is important to note that this dataset exhibits significant class imbalance, with a large number of roles having only a few lines while a few roles have a substantial number of lines, which is a good fit for large-scale non-i.i.d. settings.  Please kindly refer to the additional discussions on these datasets ``Section D. Additional Experimental Results'' in the appendices.
> > > > > > >
> > > > > > > [Lai et al. 2022] Fan Lai, Yinwei Dai, Xiangfeng Zhu, and Mosharaf
> > > > > > > Chowdhury. Fedscale: Benchmarking model and system performance of federated learning at scale. ICML 2022.
> > > > > > >
> > > > > > >
> > > > > > > [Li et al. 2022a] Chenning Li, Xiao Zeng, Mi Zhang, and Zhichao Cao.
> > > > > > > Pyramidfl: A fine-grained client selection framework for
> > > > > > > efficient federated learning. In ACM MobiCom, pages
> > > > > > > 158–171, 2022.
> > > > > > >
> > > > > > > [Li et al. 2022b] Xin-Chun Li, Yi-Chu Xu, Shaoming Song, Bingshuai
> > > > > > > Li, Yinchuan Li, Yunfeng Shao, and De-Chuan Zhan.
> > > > > > > Federated learning with position-aware neurons. In
> > > > > > > CVPR, pages 10082–10091, 2022
> > > > > > >
> > > > > > > [Qu et al. 2022] Liangqiong Qu, Yuyin Zhou, Paul Pu Liang, Yingda
> > > > > > > Xia, Feifei Wang, Ehsan Adeli, Li Fei-Fei, and Daniel
> > > > > > > Rubin. Rethinking architecture design for tackling data
> > > > > > > heterogeneity in federated learning. In CVPR, pages
> > > > > > > 10061–10071, 2022.
> > > > > > >
> > > > > > > [Lai et al. 2021] Fan Lai and Xiangfeng Zhu, Harsha V. Madhyastha, and
> > > > > > > Mosharaf Chowdhury. Oort: Efficient federated learning
> > > > > > > via guided participant selection. In Proc. of USENIX
> > > > > > > OSDI, pages 19–35, 2021.

---

> > > > > > > ### Author Response · Authors · 2023-08-24
> > > > > > > **Response to follow-up questions (3/3)**
> > > > > > >
> > > > > > > **Response to Question 4:** Thank you for this comment. In the following, we would like to clearly differentiate the synchronous SGD, fully asynchronous SGD, and our proposed adaptive asynchronous SGD in this paper.
> > > > > > >
> > > > > > > **Synchronous SGD:**  Synchronous SGD is a method of distributed training where all worker nodes synchronize (gossip average) after each iteration to ensure that they are always working at the same pace as updating the model. While it ensures consistency, it can be slower due to the need to wait for all workers to finish their computations.
> > > > > > >
> > > > > > > **Fully Asynchronous SGD:** The fully asynchronous SGD breaks the synchronization in synchronous SGD by allowing workers to use stale weights to compute gradients, which significantly reduces the idle time for synchronization. That being said, workers update their models according to their own paces.
> > > > > > >
> > > > > > > **DSGD-AAU (Algorithm 1):** The DSGD-AAU algorithm dynamically determines the number of fastest neighbor workers for each worker and differentiates stragglers iteration-by-iteration based on the running time. Specifically, in each iteration, only a subset of workers participate (rather than all as in synchronous SGD for decentralized learning), and each worker only waits for a subset of neighbors to perform local model updates (instead of waiting for no neighbor as in fully asynchronous SGD for decentralized learning). In other words, DSGD-AAU achieves the best-of-both-world of synchronous and fully asynchronous SGD (see the introduction for more motivations).
> > > > > > >
> > > > > > >
> > > > > > >
> > > > > > > The dynamic determination in DSGD-AAU allows it to mitigate the effect of stragglers at each iteration.
> > > > > > > As we presented in Algorithm 1 and Section 3, at each iteration $k$, the key is to determine the set of fastest workers, i.e., $\mathcal{N}(k)\in\mathcal{N}$,  which completed the local gradient computations, and only workers in set $
> > > > > > > \mathcal{N}(k)$ actively conduct gossip-average with neighbors while workers in set $\mathcal{N}\setminus\mathcal{N}(k)$ are still computing their local gradients. The iteration $k$ begins at finding the set $\mathcal{N}(k)$ and ends when $\mathcal{N}(k)$ is determined with workers inside it having completed the local gradient computations and the gossip-average operations. That being said, for workers finished the local gradient computation at iteration $k$, $\mathbf{w}(k+1)\neq \mathbf{w}(k)$, while for workers who didn't finish the local gradient computation, $\mathbf{w}(k+1)=\mathbf{w}(k)$.
> > > > > > >
> > > > > > >
> > > > > > >
> > > > > > > We thank this reviewer again for providing us a chance to further clarify the experiments in our paper. We will definitely add the above clarification to improve the presentation in the camera-ready version.

---

### Review · Reviewer_ndJA · 2023-07-25

**Summary Of Contributions:**

This paper proposes a fully decentralized algorithm DSGD-AAU with adaptive asynchronous updates via adaptively determining the number of neighbor workers for each worker to communicate with. The authors prove in theory that the proposed algorithms have linear speed up in the convergence. And empirical results are given on 2-NN, VGG, AlexNet and ResNet.

**Audience:**

Yes

**Claims And Evidence:**

Yes

**Requested Changes:**

* Could you compare DSGD-AAU to [MATCHA](https://arxiv.org/abs/1905.09435)?
* Could you elaborate how you measured iterations in theory?
* Could you elaborate more on the experimental setting (especially on the system side)? (see weaknesses)

**Strengths And Weaknesses:**

Strengths:
* The paper has good motivation to mitigate straggler issues in the asynchronous decentralized optimization problem.
* Theoretical analysis is given in this paper showing the proposed algorithm DSGD-AAU has linear speed up in convergence.

Weaknesses
* The paper proposes a technique that looks similar to [MATCHA](https://arxiv.org/abs/1905.09435). It is not clear how DSGD-AAU would compare to MATCHA in terms of straggler mitigation and convergence speed.
* It's not clear to me how authors define an iteration in theory. In the baseline papers like [AD-PSGD](https://arxiv.org/abs/1710.06952), a single gradient update on whichever worker is counted as one iteration. For asynchronous algorithm if the convergence bound shows linear speed up w.r.t. the worker numbers, it brings up confusion since it's likely some of the workers might not be chosen (e.g. an extremely slow straggler)
* The experiments are a little outdated. Models like AlexNet are not actively used today.
* It is also not clear how the authors split 128 workers to 3 GPUs, how they measured the wall-clock time effectively, and how to avoid the deadlocks in the baseline works. More explanations on these details would be beneficial.

---

> ### Author Response · Authors · 2023-08-03
> **Response to Reviewer ndJA (1/4)**
>
> Thank you for your constructive feedback and comments. Here we would like to address the reviewer’s concerns and
> our point-to-point responses are as follows:
>
> **Weakness #1:** The paper proposes a technique that looks similar to MATCHA. It is not clear how DSGD-AAU would compare to MATCHA in terms of straggler mitigation and convergence speed.
>
> **Our Response:** Thank you for providing us an opportunity to further clarify our proposed DSGD-AAU algorithm. As a state-of-the-art method, we were indeed aware of ``MATCHA''. We did not compare with MATCHA since our DSGD-AAU is fundamentally different from MATCHA from the following perspectives:
>
> **1. Straggler Mitigation:** It is known that there are several facts that can lead to stragglers in decentralized/federated learning, among which two common ones are the computational stragglers and the communication stragglers. Specifically, the computational stragglers refer to workers that cause delays due to slow computation, while the communication stragglers refer to workers that cause delays due to slow communication. As a result, these two types of stragglers are resultant from different factors, i.e., the computational stragglers could be due to the worker's hardware limitations, high CPU usage from other processes, or other computation-related issues, while the communication stragglers could be due to network congestion, poor network connectivity, or other network-related issues. In this paper, we mainly focus on the computational stragglers, while MATCHA mainly focuses on the communication stragglers. As a result, our DSGD-AAU and the state-of-the-art MATCHA are fundamentally different and are designed to address different types of stragglers in decentralized/federated learning.
>
> **2. Pathsearch vs. Link Control:** The second major difference lies in how to manipulate the topology of all workers. Since DSGD-AAU is designed specifically to mitigate the effect of computational stragglers, it does this by adaptively determining the number of neighbor workers for each worker to perform local model updates during the training process. This means that DSGD-AAU dynamically selects the fastest computational workers to participate in each iteration, which reduces the impact of slower workers (computational stragglers) on the overall process. While MATCHA does not explicitly focus on communication straggler mitigation, it does aim to optimize the overall runtime by carefully tuning the frequency of inter-worker communication. This could indirectly address communication straggler issues by ensuring that the communication is more frequent over critical links, potentially reducing the impact of slower communicational workers on the overall process. Due to the different properties of computational stragglers and communication stragglers, our DSGD-AAU updates in an adaptive asynchronous manner, i.e., workers update the local parameters asynchronously, while the link sampling in MATCHA still makes workers update local parameters in a synchronous manner.
>
> **3. Theoretical Aspect on Convergence Speed:** Although both our DSGD-AAU and MATCHA provide convergence analysis showing that DSGD-AAU and MATCHA achieve  a linear speedup of convergence in terms of iterations with respect to the total number of workers for a sufficiently large number of communication iterations, the convergence in wall-clock time is significantly different for these two algorithms, since the delays in DSGD-AAU are caused by computational stragglers and the delays in MATCH are caused by the communication stragglers. In addition, our proofs for showing convergence in terms of iteration are different. MATCHA leverages the conventional spectral gap  and spectral norms while DSGD-AAU uses the bounded connectivity-time in Assumption 4.2.
>
> Nevertheless, in decentralized/federated learning, both types of stragglers can significantly impact the speed and efficiency of the training process. Therefore, different strategies and algorithms, such as DSGD-AAU and MATCHA, are designed to mitigate the effects of these stragglers. In summary, while both algorithms aim to improve the efficiency of decentralized SGD, they do so in two different ways and are particularly effective in different scenarios. DSGD-AAU is more focused on mitigating the impact of computational stragglers and can achieve faster convergence when the computational capabilities of the workers vary significantly. On the other hand, MATCHA is more focused on optimizing inter-node communication and can achieve faster convergence when the communication delays between different nodes vary significantly.

---

> > ### Author Response · Authors · 2023-08-03
> > **Response to Reviewer ndJA (2/4)**
> >
> > **Weakness #2:** It's not clear to me how authors define an iteration in theory. In the baseline papers like AD-PSGD, a single gradient update on whichever worker is counted as one iteration. For asynchronous algorithm if the convergence bound shows linear speed up w.r.t. the worker numbers, it brings up confusion since it's likely some of the workers might not be chosen (e.g. an extremely slow straggler).
> >
> > **Our Response:** Thank you for this insightful comment, and indeed it is very critical to properly define an iteration for non-synchronous algorithms. As we presented in Algorithm 1 and Section 3, at each iteration $k$, the key is to determine the set of fastest workers, i.e., $\mathcal{N}(k)\in\mathcal{N}$,  which completed the local gradient computations, and only workers in set $
> > \mathcal{N}(k)$ actively conduct gossip-average with neighbors while workers in set $\mathcal{N}\setminus\mathcal{N}(k)$ are still computing their local gradients. The iteration $k$ begins at finding the set $\mathcal{N}(k)$ and ends when $\mathcal{N}(k)$ is determined with workers inside it having completed the local gradient computations and the gossip-average operations. Based on this general definition of $k$, we provide the theoretical analysis in Section 4 and propose a novel implementation so as to find the set of workers in $\mathcal{N}(k)$ to realize Algorithm 1. As a result, one of the major contributions in this paper is to propose the Pathsearch procedure (see Section 5, and Section B ``The Proposed Path Searching Procedure'' in the appendices), which is essentially used to determine when a logical iteration ends.
> >
> > Specifically, at each iteration $k$, a worker $j_k$ that completes the local gradient computation first performs a local SGD step. This is followed by a step of the graph searching procedure to establish a new edge in the set $\mathcal{P}$. The next operation in the current iteration is the gossip-average of the local parameters of workers $j_k$ in the set $\mathcal{N}\_{j_k}(k)$. If any other worker $i_k$ finishes the local gradient update during the Pathsearch of the fast worker $j_k$, it will also run Pathsearch independently until $\mathcal{P}$ and $\mathcal{V}$ get updated. Hence, worker $i_k$ also conducts the gossip-average of the local parameters in the set $\mathcal{N}_{i_k}(k)$. The Pathsearch procedure ends at each iteration only when a new link is established in $\mathcal{P}$. This definition of an iteration in DSGD-AAU is different from the one in AD-PSGD where a single gradient update on whichever worker is counted as one iteration. In DSGD-AAU, an iteration involves a more complex process that includes a local SGD step, a graph searching procedure, and a gossip-average operation. Note that the iteration defined in DSGD-AAU is fundamentally different from that defined in conventional fully asynchronous updates such as the AD-PSGD in Lian et. al., 2018, where an iteration ends whenever a worker completes the local computation. The deficiency of the conventionally defined iteration often causes the ``stale information issue'', i.e., the parameter updates of workers with fast computation capabilities will be dragged down by the stale information of workers with slow computation abilities, as shown in Figure. 1(b). Hence, the convergence of AD-PSGD in Lian et al. 2018 is established under the assumption that the staleness must be bounded, while the algorithm itself cannot guarantee such a bound.
> >
> > Regarding your concern on the linear speedup with respect to the number of workers, it is important to note that DSGD-AAU adaptively determines the number of neighbor workers for each worker to perform local model updates during the training time. This means that the algorithm dynamically selects the fastest computational workers to participate in each iteration, which reduces the impact of slower workers (stragglers) on the overall process. Moreover, since Pathsearch guarantees that a new worker is involved in the parameter exchange at each iteration, and all {workers} will participate at least once every $N-1$ iteration. This is also a key property of the proposed DSGD-AAU.

---

> > > ### Author Response · Authors · 2023-08-03
> > > **Response to Reviewer ndJA (3/4)**
> > >
> > > **Weakness #3:** The experiments are a little outdated. Models like AlexNet are not actively used today.
> > >
> > > **Our Response:** Thank you for this comment. Note that we considered **four representative DNN models** (2NN, AlexNet, VGG-13 and ResNet-18) over two tasks (image classification and natural language processing) using several widely used datasets. To the best of our knowledge, these DNN models, e.g., VGG, ResNet have been extensively used in the evaluations of existing works on decentralized/federated learning, for instance [Hsieh et al. 2020, Assran et al. 2020, Luo et al. 2020, Nadiradze et al. 2021, Yang et al. 2021, Wang et al. 2022], just to name a few.
> > >
> > > [Hsieh et al. 2020] Kevin Hsieh, Amar Phanishayee, Onur Mutlu, and Phillip Gibbons. The Non-IID Data Quagmire of Decentralized Machine Learning. In Proc. of ICML, 2020.
> > >
> > > [Assran et al. 2020] Mahmoud Assran, Arda Aytekin, Hamid Reza Feyzmahdavian, Mikael Johansson, and Michael G Rabbat. Advances in asynchronous parallel and distributed optimization. Proceedings of the IEEE, 108(11):2013–2031, 2020.
> > >
> > > [Luo et al. 2020] Qinyi Luo, Jiaao He, Youwei Zhuo, and Xuehai Qian. Prague: High-Performance Heterogeneity-Aware Asynchronous Decentralized Training. In Proc. of ACM ASPLOS, 2020.
> > >
> > > [Nadiradze et al. 2021] Giorgi Nadiradze, Amirmojtaba Sabour, Peter Davies, Shigang Li, and Dan Alistarh. Asynchronous decentralized sgd with quantized and local updates. Advances in Neural Information Processing Systems, 34:6829–6842, 2021.
> > >
> > > [Yang et al. 2021] Haibo Yang, Minghong Fang, and Jia Liu. Achieving Linear Speedup with Partial Worker Participation in Non-IID Federated Learning. In Proc. of ICLR, 2021.
> > >
> > > [Wang et al. 2022] Haoxiang Wang, Zhanhong Jiang, Chao Liu, Soumik Sarkar, Dongxiang Jiang, and Young M Lee. Asyn- chronous training schemes in distributed learning with time delay. arXiv preprint arXiv:2208.13154, 2022.
> > >
> > > **Weakness #4:** It is also not clear how the authors split 128 workers to 3 GPUs, how they measured the wall-clock time effectively, and how to avoid the deadlocks in the baseline works. More explanations on these details would be beneficial.
> > >
> > > **Our Response:** Thank you for this comment and providing us an opportunity for further clarifying our experiments. In particular, we randomly assign all workers to 3 GPUs, each of which with the same configurations and computational capability. All workers operate in parallel and after completing one iteration, the worker records its current timestamp and model. One shared file is used to record each worker's local training process, which can be accessed by all workers through the NFS (Network File System). During the Pathsearch procedure, if one worker is selected, then it steps into the local model update process and also records the current timestamp (system time). By comparing the two timestamps (initial and current), the wall-clock time can be computed easily for each worker. A deadlock arises when multiple communications wait for each other in a cyclic manner. Specifically, when a communication is in need of a resource locked by another communication, it remains in a waiting state until either the lock is granted or a timeout period elapses. To address this issue, we leverage the timeout method, which determines the duration a communication that is willing to wait before abandoning its attempt to acquire the resource. The timeout method has been extensively utilized in FedScale [Lai et al. 2022] and other related works [Li et al. 2022a, Li et al. 2022b, Qu et al. 2022, Lai et al. 2021]  to prevent prolonged and unnecessary waiting.
> > >
> > > [Lai et al. 2022] Fan Lai, Yinwei Dai, Xiangfeng Zhu, and Mosharaf Chowdhury. Fedscale: Benchmarking model and system performance of federated learning at scale. ICML 2022.
> > >
> > > [Li et al. 2022a] Chenning Li, Xiao Zeng, Mi Zhang, and Zhichao Cao. Pyramidfl: A fine-grained client selection framework for
> > > efficient federated learning. In ACM MobiCom, pages 158–171, 2022.
> > >
> > > [Li et al. 2022b] Xin-Chun Li, Yi-Chu Xu, Shaoming Song, Bingshuai Li, Yinchuan Li, Yunfeng Shao, and De-Chuan Zhan. Federated learning with position-aware neurons. In CVPR, pages 10082–10091, 2022
> > >
> > > [Qu et al. 2022] Liangqiong Qu, Yuyin Zhou, Paul Pu Liang, Yingda Xia, Feifei Wang, Ehsan Adeli, Li Fei-Fei, and Daniel
> > > Rubin. Rethinking architecture design for tackling data heterogeneity in federated learning. In CVPR, pages 10061–10071, 2022.
> > >
> > > [Lai et al. 2021] Fan Lai and Xiangfeng Zhu, Harsha V. Madhyastha, and Mosharaf Chowdhury. Oort: Efficient federated learning
> > > via guided participant selection. In Proc. of USENIX OSDI, pages 19–35, 2021.

---

> > > > ### Author Response · Authors · 2023-08-03
> > > > **Response to Reviewer ndJA (4/4)**
> > > >
> > > > **Requested Change #1:** Could you compare DSGD-AAU to MATCHA?
> > > >
> > > > **Our Response:** Please kindly refer to **our  response to Weakness #1**. To make it clear, we add a discussion in Section 7 as "Although we focus on stragglers caused by heterogeneous computation capabilities among workers as in previous work in (Ananthanarayanan et al., 2013; Ho et al., 2013; Lian et al., 2017; 2018; Luo et al., 2020), there is another line of works focusing on another key type of stragglers, i.e., the communication stragglers Wang et al. (2019); Wang & Joshi (2021); Wang et al. (2020b). In particular, the celebrated  ``MATCHA'' algorithm was proposed  in Wang et al. (2019)  to mitigate the communication stragglers. Note that though MATCHA also has a link-control procedure,  it differentiates from our proposed DSGD-AAU algorithm fundamentally. Specifically, DSGD-AAU adaptively
> > > > selects the fastest computational workers for each iteration to mitigate the effect of slower computational workers, while
> > > > MATCHA optimizes runtime by tuning the frequency of inter-node communication, potentially reducing the impact of slower communication workers, which leads DSGD-AAU to be an asynchronous algorithm and MACTHA to be a synchronous one. Furthermore, in contrast to MATCHA, our proposed DSGD-AAU leverages the bounded-connectivity time to theoretically show the convergence rather than using the conventional spectral radius as in Lian et al. (2017; 2018). These differences highlight that DSGD-AAU and MATCHA are designed to address distinct challenges in the field."
> > > >
> > > > **Requested Change #2:**  Could you elaborate how you measured iterations in theory?
> > > >
> > > > **Our Response:** Please kindly refer to **our response to Weakness #2**. To make it clear, we add the following sentences in Section 3 as "Note that at each iteration $k$ in Algorithm 1, the key is to determine the set of fastest workers, i.e., $\mathcal{N}(k)\in\mathcal{N}$,  which have completed the local gradient computations, and only workers in set $\mathcal{N}(k)$ actively conduct gossip-average with neighbors while workers in set $\mathcal{N}\setminus\mathcal{N}(k)$ are still computing their local gradients. The iteration $k$ begins at finding the set $\mathcal{N}(k)$ and ends when $\mathcal{N}(k)$ is determined with workers inside it having completed the local gradient computations and the gossip-average operations.'' In addition, we modify Remark 5.1 with additional discussions as follows  ``One of our key contributions is the Pathsearch procedure, which is essentially used to determine when a logical iteration ends. The Pathsearch procedure ends at each iteration only when a new link is established in $\mathcal{P}$. This definition of an iteration in DSGD-AAU is different from the one in AD-PSGD where a single gradient update on whichever worker is counted as one iteration. In DSGD-AAU, an iteration involves a more complex process that includes a local SGD step, a graph searching procedure, and a gossip-average operation. Moreover, since Pathsearch guarantees at each iteration, a new worker must be involved in the parameter exchange, and all workers will participate at least once every $N-1$ iterations. This is also a key property of our proposed DSGD-AAU.''
> > > >
> > > >
> > > >
> > > > **Requested Change #3:** Could you elaborate more on the experimental setting (especially on the system side)? (see weaknesses)
> > > >
> > > > **Our Response:** Please kindly refer to **our response to Weakness #4**. Indeed, we provided some additional discussions on experimental settings in Section D "Additional Experimental Results" in the appendices. Following the reviewer's suggestions and based on our earlier response, we add the following discussions to Section 6 as ``In real systems, there are several reasons leading to the existing of stragglers, e.g., heterogeneous hardware, hardware failure, imbalanced data distributions among tasks and different OS effects, resource contention, etc. Since we run our experiments of different number of workers in one sever as in many other works, e.g., Li et al. 2020 a,b and motivated by the idea of AD-PSGD Lian et. al 2018 and Cipar et al. 2013, we randomly select workers as stragglers in each iteration.  If a worker is selected to be a straggler, then it will sleep for some time in the iteration (e.g., the sleep time could be 6x of the average one local computation time).  We also investigate the impact of the sleep time and the percentage of stragglers in our ablation study, see Section D. Such techniques have been widely used in the literature such as AD-PSGD (Lian et al. 2018), Prague (Luo et al. 2020) and Li et al. 2020 a,b.''

---

### Decision · Action_Editors · 2023-09-26

**Recommendation:** Reject

**Comment:**

This work proposes a method for decentralized optimization in distributed networks which aims to mitigate the impact of (computational) stragglers. While all reviewers agreed that this is an important and interesting direction of work, there were shared concerns regarding the clarity of the work, empirical setup, connections to prior work, and gaps between theory/practice. The authors carefully considered these points with their rebuttal, but several key concerns (regarding the clarity of the paper, empirical settings, and divides between theory/practice) remained after the discussion. We believe the work is encouraging and could lead to a strong revised resubmission if the authors can address/incorporate these points.

**Audience:**

This general area that this work explores (decentralized optimization of ML objectives) is of interest to the TMLR community.

**Claims And Evidence:**

All reviewers agreed that this work is interesting and aims to tackle an important problem. However, there were shared concerns around the clarity of the work and the empirical/theoretical results, particularly given potential gaps between the theoretical and empirical results, and the scope of the experiments. In particular, reviewers agreed that the work could benefit from a revision to consider experiments on more recent workloads (e.g., larger models and larger-scale distributed networks), as well as carefully incorporating the discussions from the rebuttal period to improve the clarity of the work and better highlight the novel aspects relative to existing work in decentralized optimization.

**Resubmission Of Major Revision:**

The authors may consider submitting a major revision at a later time.